# Genetic modification of primary human B cells to model high-grade lymphoma

Rebecca Caeser[1,2], Miriam Di Re[1,2], Joanna A. Krupka [1,2,3], Jie Gao[1,2], Maribel Lara-Chica [4], João M.L. Dias [4], Susanna L. Cooke[5], Rachel Fenner[1,2], Zelvera Usheva[1,2], Hendrik F.P. Runge [1,2], Philip A. Beer[6], Hesham Eldaly[7,8], Hyo-Kyung Pak[9], Chan-Sik Park[9], George S. Vassiliou [1,2,6], Brian J.P. Huntly [1,2], Annalisa Mupo[4], Rachael J.M. Bashford-Rogers [10] & Daniel J. Hodson [1,2]*

Sequencing studies of diffuse large B cell lymphoma (DLBCL) have identified hundreds of recurrently altered genes. However, it remains largely unknown whether and how these mutations may contribute to lymphomagenesis, either individually or in combination. Existing strategies to address this problem predominantly utilize cell lines, which are limited by their initial characteristics and subsequent adaptions to prolonged in vitro culture. Here, we describe a co-culture system that enables the ex vivo expansion and viral transduction of primary human germinal center B cells. Incorporation of CRISPR/Cas9 technology enables high-throughput functional interrogation of genes recurrently mutated in DLBCL. Using a backbone of *BCL2* with either *BCL6* or *MYC*, we identify co-operating genetic alterations that promote growth or even full transformation into synthetically engineered DLBCL models. The resulting tumors can be expanded and sequentially transplanted in vivo, providing a scalable platform to test putative cancer genes and to create mutation-directed, bespoke lymphoma models.

[1] Wellcome MRC Cambridge Stem Cell Institute, Cambridge CB2 0AW, UK. [2] Department of Haematology, University of Cambridge, Cambridge, UK. [3] MRC Cancer Unit, University of Cambridge, Hutchison/MRC Research Centre, Cambridge, UK. [4] Cancer Molecular Diagnostics Laboratory (CMDL), Department of Haematology, University of Cambridge, Cambridge, UK. [5] Wolfson Wohl Cancer Research Centre, Institute of Cancer Sciences, University of Glasgow, Garscube Estate, Glasgow, UK. [6] Wellcome Sanger Institute, Wellcome Genome Campus, Hinxton, CA CB10 1SA, UK. [7] Department of Pathology, Cambridge University Hospitals, Cambridge, UK. [8] Department of Clinical Pathology, Cairo University, Giza, Egypt. [9] Department of Pathology, University of Ulsan College of Medicine, Asan Medical Centre, Seoul, Korea. [10] Wellcome Centre for Human Genetics, Roosevelt Dr, Oxford OX3 7BN, UK. *email: djh1002@cam.ac.uk

Diffuse large B cell lymphoma (DLBCL) is the most common form of non-Hodgkin lymphoma. Although potentially curable with immunochemotherapy, up to 40% of patients succumb to their disease[1]. In an attempt to unravel the biological basis of DLBCL and to identify new therapeutic opportunities, several groups have recently reported large genomic studies[2–4]. These highlight the considerable genetic heterogeneity of DLBCL and identify hundreds of recurrently mutated genes, copy number alterations, and structural variants. Clusters of co-mutated genes suggest the existence of genetic subtypes of DLBCL that may behave differently when exposed to therapeutic agents. While the functional and mechanistic consequences of some of these genetic alterations have been established, for the majority we have little to no understanding of their contribution to lymphomagenesis. To translate these genomic findings into therapeutic progress, it is critical to understand the functional importance and therapeutic relevance of these genetic alterations, both individually and in combination.

Existing model systems used for the functional interrogation of lymphoma genetics consist predominantly of lymphoma cell lines and genetically modified mice. However, both have limitations; cell lines were often established from patients with end-stage, non-nodal or even leukemic phase lymphoma and carry an extensive and biased mutational repertoire, further selected over years or even decades of in vitro growth. Genetically engineered mice, on the other hand, are costly, time-consuming to generate, and therefore unsuitable for high-throughput or combinatorial experiments. Furthermore, the genetic requirements for tumorigenesis in mice do not always accurately reflect those in humans[5]. As such, the development of new, preclinical models of lymphoma that can capture its considerable genetic diversity has been identified as a priority area for lymphoma research[6].

In common with many of the mature B cell malignancies, DLBCL is thought to arise from the germinal center (GC) stage of B cell differentiation[7,8]. An attractive solution would therefore be to use primary human GC B cells as a platform for ex vivo genetic manipulation. Equivalent approaches have proved fruitful for epithelial malignancies. However, technical difficulties associated with the ex vivo culture and genetic manipulation of human GC B cells, including high manipulation-associated cell toxicity and low transduction efficiency, have obstructed the exploitation of such models to study lymphoma.

Here, we describe an optimized strategy that facilitates proliferation and highly efficient transduction of non-malignant, primary, human GC B cells ex vivo. We show that combinations of oncogenes permit long-term culture in vitro, allowing the system to be used for high-throughput screening of oncogenes and tumor suppressors, and for the creation of genetically customized human lymphoma models that can be studied in immunodeficient mice.

## Results

### Ex vivo growth and transduction of primary human GC B cells.
GC B cells are programmed to undergo apoptosis in the absence of survival signals from T follicular helper cells and follicular dendritic cells (FDC). Consistent with this, it is well-established that GC B cells perish rapidly if cultured unsupported ex vivo[9]. Previous attempts to support ex vivo growth of human GC B cells employed CD40 ligand (CD40lg)-transfected fibroblasts in combination with soluble cytokines including interleukin2 (IL2), IL4, and IL10[9,10]. Related strategies have used an FDC-like feeder cell, termed HK, that supported GC survival and allowed short-term proliferation when combined with CD40lg[11]. With the increasing appreciation of the importance of IL21 to GC B cell biology[12,13], later systems have used HK feeder cells combined with CD40lg

and IL21 (ref. [14]). However, proliferation of GC B cells in all these systems was typically limited to a period of up to 10 days[9–11,14].

We employed a similar system based upon a freshly established culture of modified HK cells, termed YK6 that were immortalized with TERT, P53dd, and CDK4 (Supplementary Fig. 1a). Initial experiments suggested that membrane-expressed CD40lg in combination with IL21 facilitated robust stimulation of GC B cells (Supplementary Fig. 1b). We therefore engineered our immortalized YK6 cells to express membrane human CD40lg and to secrete soluble IL21, termed YK6-CD40lg-IL21 (Supplementary Fig. 1c). We isolated primary GC B cells (CD38+CD20+CD19+CD10+) from pediatric tonsil tissue (Fig. 1a), which when grown in co-culture with YK6-CD40lg-IL21 survived and proliferated vigorously for up to 10 days without a requirement for any additional cytokines (Fig. 1b, c, Supplementary Movies 1–5).

In line with previous observations in human B cells[15,16] we were unable to transduce human GC B cells with amphotrophic or VSV-G pseudotyped virus. Peripheral blood B cells have previously been transduced using virus pseudotyped with a Gibbon Ape Leukemia Virus (GaLV) envelope[17], the receptor for which is *SLC20A1* (ref. [18]). RNA-Seq showed that human GC B cells express high levels of *SLC20A1*, but very low levels of the VSV-G receptor *LDLR* (Fig. 1d). Thus, we proceeded to test the GaLV viral envelope to transduce primary GC B cells. To permit lentiviral transduction, we generated a series of GaLV-MuLV fusion constructs based on previous reports[17,19] (Fig. 1e) and identified a fusion construct that permitted high efficiency transduction with both retroviral (Fig. 1f) and lentiviral (Fig. 1g) constructs of human primary GC B cells cultured on YK6-CD40lg-IL21 feeders. Interestingly, the GaLV envelopes also enabled the transduction of primary human DLBCL cells supported on YK6-CD40lg-IL21 cells (Supplementary Fig. 1d).

### Long-term expansion of human GC B cells ex vivo.
We proceeded to use this culture-transduction system to introduce into human GC B cells oncogenes that are commonly deregulated in human lymphoma. Out of five genes tested, no single gene was able to prolong the survival of primary GC B cells cultured in our system (Fig. 2a, b). However, *BCL2* when co-expressed with either *MYC* or *BCL6* overexpression did lead to long-term expansion and survival of transduced GC B cells in culture. These cells continued to expand and proliferate vigorously in culture beyond 100 days. We also tested other transcription factors associated with the GC reaction, and their lymphoma-associated mutants, in combination with BCL2 in a pooled, competitive culture. This showed initial expansion of cells transduced with *MEF2B* Y69H, a mutation commonly found in DLBCL and follicular lymphoma[20]. However, by day 59, cultures were dominated by *BCL6*-transduced cells suggesting this as the transcription factor best able to promote long-term growth of GC B cells ex vivo (Fig. 2c, Supplementary Data 1). Flow cytometry after 10 weeks of culture showed that cells transduced with *BCL2* and *BCL6* maintained expression of surface markers reminiscent of GC B cells including CD19, CD20, CD22, CD38, CD80, and CD95 (Fig. 2d). Cells expressed both CD86 and CXCR4 markers, an immunophenotype intermediate between light and dark zone GC B cells (Fig. 2d). Cells transduced with *BCL2* and *MYC* remained viable and proliferated but downregulated CD20 and CD19, consistent with differentiation towards plasmablasts (Supplementary Fig. 1e). The plasma cell marker CD138 was not expressed by either *BCL2*/*MYC* or *BCL2*/*BCL6* transduced cells (Supplementary Fig. 1f). We compared gene expression profiles of freshly isolated and transduced GC B cells cultured ex vivo at early (5 days) and late (10 weeks) time points (Fig. 2e, Supplementary

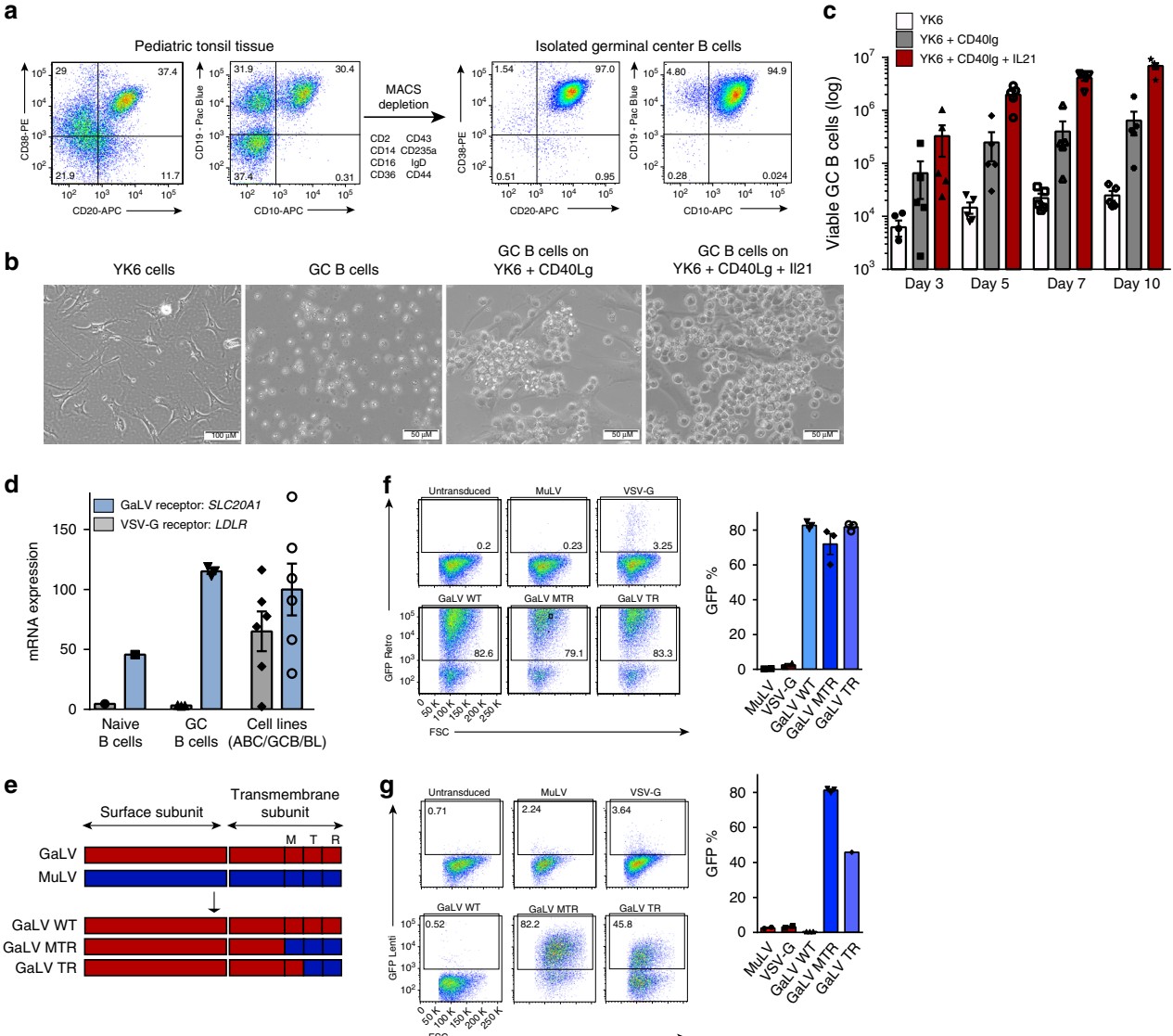

**Fig. 1** Ex vivo growth and transduction of primary human GC B cells. **a** Representative flow cytometry analysis (n > 3) for the expression of GC B cell markers CD38, CD20, CD19, and CD10 in purified GC B cells from pediatric tonsil tissue. The strategy for negative selection of GC B cells is shown. **b** Representative images are shown of YK6 cells, GC B cells alone, or GC B cells cultured on either YK6-CD40lg or YK6-CD40lg-IL21 feeder cells. Scale bar represents 50 or 100 μm as indicated. Source data are provided as a Source Data file. **c** Primary human GC B cells were cultured with YK6 control, YK6-CD40lg, or YK6-CD40lg-IL21 feeder cells. Illustrated is bar graph showing the number of viable cells ( ± s.e.m., n = 5) over four timepoints. Viable cells were determined by flow cytometry and counting beads. Source data are provided as a Source Data file. **d** Bar graph showing the relative transcript expression of *SLC20A1* (GaLV receptor) and *LDLR* (VSV-G receptor) in naïve (n = 1), GC B cells (n = 3), and ABC/GCB DLBCL and Burkitt cell lines (n = 6) as analyzed by RNA-seq. mRNA expression values were calculated as counts per million reads (CPM). Error bars indicate ± s.e.m. Source data are provided as a Source Data file. **e** Schematic of the retroviral and lentiviral MuLV-GaLV fusion envelopes, GaLV_WT, GaLV_MTR, and GaLV_TR. M = transmembrane region, T = cytoplasmic tail, R = R peptide, SU = surface subunit, TM = transmembrane subunit[19]. **f, g** Primary human GC B cells were transduced with a retroviral control (**f**) or lentiviral control (**g**) construct using GaLV-MuLV fusion envelope constructs as well as VSV-G and MuLV. Three days after transduction, transduction efficiencies in primary human GC B cells were determined by expression of GFP. Error bars indicate ± s.e.m., n = 3. FSC forward scatter. Source data are provided as a Source Data file

Table 1). As anticipated, this showed enrichment of a STAT3-signature in cultured cells consistent with ongoing IL21 stimulation. While freshly isolated GC B cells were enriched for expression of centroblast genes, the cultured and transduced cells adopted a gene expression profile more similar to that of centrocytes, consistent with ongoing CD40 stimulation. Importantly, the centrocyte is the stage of GC differentiation most similar to DLBCL[21]. Transcriptome analysis was also compared with that of six cell lines commonly used as models of GC-derived lymphomas, including the main subtypes of DLBCL and Burkitt

lymphoma. When compared to a signature of GC-expressed genes (GCB-1)[22], long-term *BCL6*-transduced cells clustered more closely with GC B cells than did the cell lines (Fig. 2f, Supplementary Fig. 1g).

Overall, these results suggest that transduced primary human GC B cells can be cultured long-term ex vivo, retaining characteristics of the initial GC B cell that are shared with DLBCL cells. This represents a valuable model system for the functional interrogation of genes involved in GC lymphomagenesis.

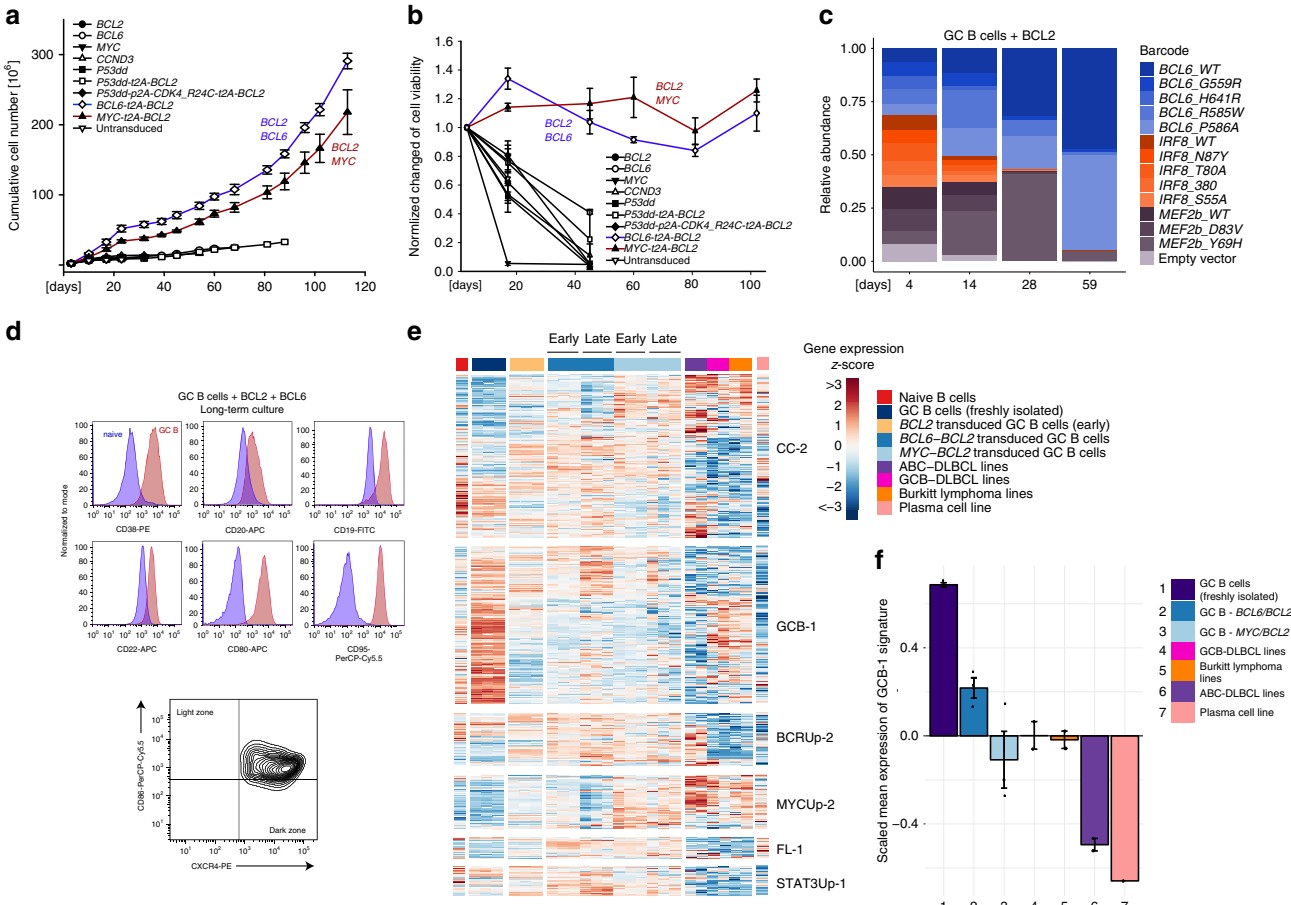

**Fig. 2** Long-term expansion of human germinal center B cells ex vivo. **a** Primary human GC B cells were transduced with the indicated oncogenes and oncogene combinations and cultured separately for up to 120 days. Graph shows calculated theoretical absolute cell numbers (±s.e.m., $n = 3$). Viable cells were assessed by trypan blue exclusion. Source data are provided as a Source Data file. **b** Primary human GC B cells were transduced with different oncogenes and oncogene combinations and monitored by flow cytometry. Graph shows the change in cell viability assessed by scatter characteristic by flow cytometry ($\pm$ s.e.m., $n = 3$). Source data are provided as a Source Data file. **c** Primary human GC B cells were transduced with BCL2 in combination with other transcription factors in a pooled, competitive culture. Graph shows relative abundance of transcription factors or their mutant versions over four different timepoints ($n = 3$). **d** Primary human GC B cells were transduced with the oncogenic cocktail BCL2 and BCL6 and cultured to day 73. Representative flow cytometry analysis ($n = 3$) for the expression of the GC B cell markers CD38, CD20, CD19, CD80, CD22, CD95, CXCR4, and CD86. Red histograms show GC B cells compared to primary human naïve B cells (blue). **e** Heat map of gene expression of freshly isolated GC B cells ($n = 3$), transduced GC B cells (BCL2-BCL6, BCL2-MYC) cultured ex vivo for 5 or 73 days ($n = 3$), plasma cell line ($n = 1$), naïve B cells ($n = 1$), and lymphoma cell lines (TMD8, HBL1, SUDHL4, DOHH2, Mutu, and Raji, $n = 6$). Illustrated are the selected gene expression signatures—CC-2, GCB-1 as well as BCRUp-2, MYCUp-2, FL-1, and STAT3Up-1 (Supplementary Table 1). Order of genes in the signatures was determined by hierarchical clustering. **f** Bar chart showing the scaled mean expression of the germinal center B cell signature, GCB-1 in freshly isolated GC B cells ($n = 3$), transduced GC B cells (BCL2-BCL6, BCL2-MYC) cultured ex vivo for 73 days ($n = 3$), plasma cell line ($n = 1$), and lymphoma cell lines (TMD8, HBL1, SUDHL4, DOHH2, Mutu and Raji, $n = 6$). Error bars represent the standard error of the mean from independent biological replicates

**Screening putative tumor suppressor genes in GC B cells.** We wished to use the system for the high-throughput study of putative tumor suppressor genes (TSGs) in lymphoma. We hypothesized that many tumor suppressor pathways are already inactivated in lymphoma cell lines, and as such, primary GC B cells should be a more sensitive platform to identify a competitive growth or survival advantage following TSG inactivation. Robust expression of Cas9 was achieved using a stable Cas9 retroviral packaging line (Supplementary Fig. 2a) and initial experiments confirmed efficient gRNA-directed targeting in primary, human, GC B cells ex vivo (Supplementary Fig. 2b, c). We therefore created a lymphoma-focused CRISPR gRNA library composed of 6000 gRNAs targeting a total of 692 genes reported to be mutated or deleted in human lymphoma, along with 250 non-targeting control guides. Each gene was targeted by up to nine gRNAs (Supplementary Fig. 2d) and deep sequencing revealed that 99%

of gRNAs were within four times of the mean frequency (Fig. 3a). The library was transduced into primary GC cells shortly after their transduction with BCL2, BCL6, and Cas9 cDNAs (experimental scheme of the CRISPR screening shown in Fig. 3b). Cas9 and gRNA constructs were marked with fluorescent proteins to allow selection to be visualized by FACS. While Cas9 and gRNA dual infected cells comprised only 10% of all cells at day 4, this population expanded to 90% by day 88 of culture (Supplementary Fig. 2e), suggesting strong selection for one or more of the library gRNAs. Genomic DNA was sequenced at intervals and a CRISPR gene score was generated for each gene (Fig. 3b).

Genes that showed the greatest enrichment during culture over 10 weeks included well-established tumor suppressors such as TP53, CDKN2A, and PTEN (Fig. 3c), thus validating the ability of our system to detect bona fide TSGs. However, the greatest enrichment was seen for GNA13 (Fig. 3c), which encodes the G

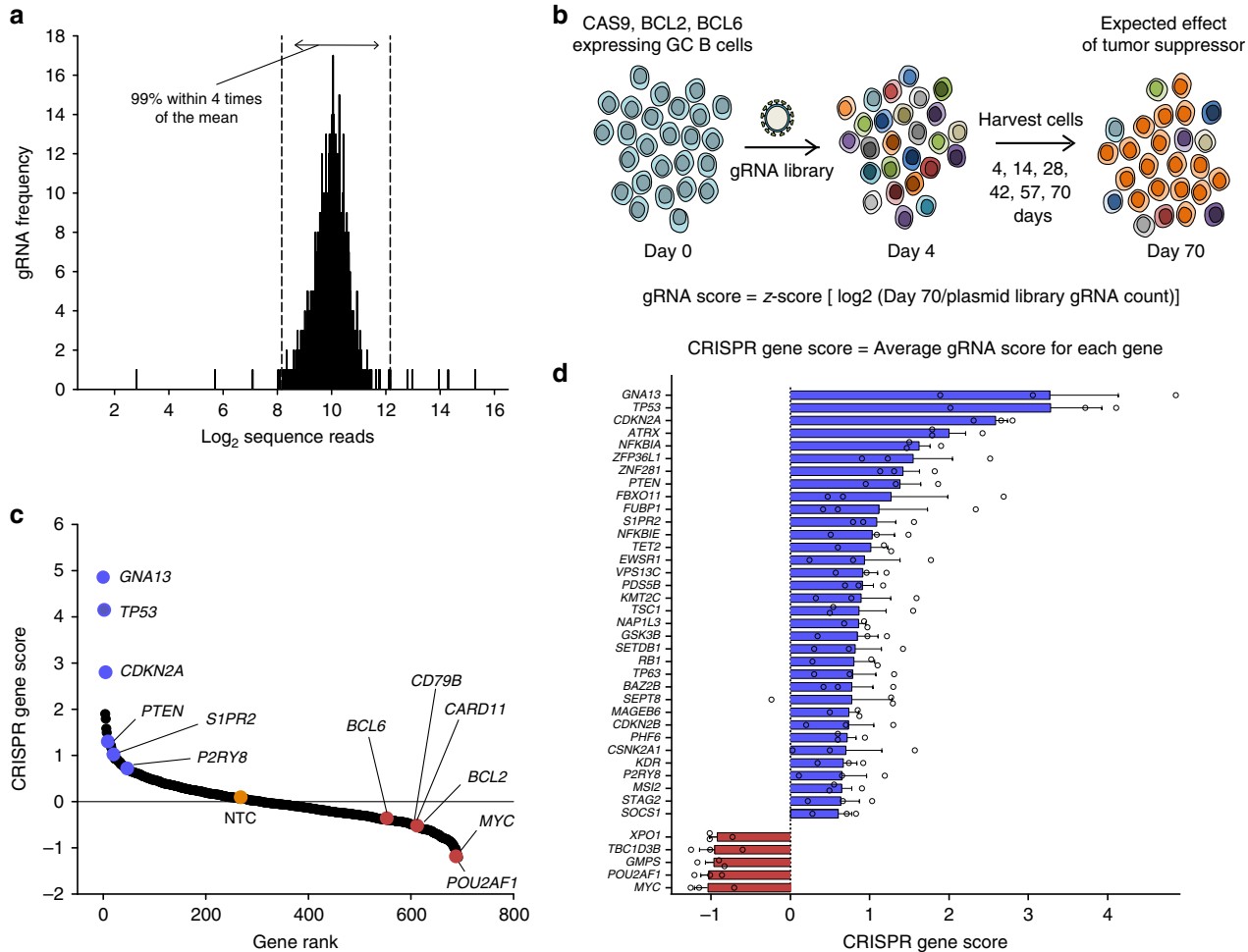

**Fig. 3** Screening putative tumor suppressor genes in human primary GC B cells. **a** Illumina sequencing of the lymphoma-focused CRISPR library revealed that 99% of sequence reads were represented within four times of the mean. Source data are provided as a Source Data file. **b** Outline of experimental design and mathematical formulas used. **c** Rank-ordered depiction of CRISPR gene scores at day 70 (log2 scale) from highest to lowest. Selected tumor suppressor genes and oncogenes are highlighted in blue and red, respectively. CRISPR gene score for non-targeting control (NTC) ($n = 250$) is 0.22. Representative of three experiments. **d** Bar graph illustrates CRISPR gene scores (log2 scale) for the top 34 enriched genes and 5 most depleted genes summarized from three independent experiments performed in separate donors. Error bars indicate ± s.e.m. for the three replicate screens. Source data are provided as a Source Data file

protein subunit α13. Indeed, the nine gRNAs targeting *GNA13* collectively represented 79% of all reads by day 70 (Supplementary Fig. 3a). Enrichment for *GNA13* was also seen in two further replicate screens performed in separate tonsil donors (Supplementary Data 2). We saw remarkable consistency of enriched genes across the three replicate screens, across which the twelve most enriched genes were *GNA13, TP53, CDKN2A, ATRX, NFKBIA, ZFP36L1, ZNF281, PTEN, FBXO11, FUBP1, S1PR2*, and *NFKBIE* (Fig. 3d).

To determine whether the oncogenic backbone used would influence the co-operating TSGs enriched, we repeated the CRISPR screen using primary GC B cells transduced with *BCL2* and *MYC*. Interestingly, a different profile of TSGs was enriched on this genetic background with much weaker enrichment of *GNA13* (Supplementary Fig. 3b, Supplementary Data 2). Instead, some of the most enriched gRNAs in the context of MYC overexpression targeted members of the ZFP36 family of RNA-binding proteins, previously demonstrated to oppose cellular transformation in a mouse model of MYC-induced lymphoma[23].

To compare the ability of our culture system to identify TSGs to that of established cell lines, we performed a parallel screen using the lymphoma cell line HBL1 (Supplementary Fig. 3c,

Supplementary Data 2) and compared data from recent published CRISPR screens[24] (Supplementary Fig. 3d). In these cell line experiments, enrichment of gRNAs targeting TSGs was much more modest. This highlights the unique potential of our primary GC culture system to identify genetic changes associated with enhanced growth and survival; a phenotype that is difficult to identify using heavily mutated cell lines, already optimized for in vitro growth.

**GNA13 depletion enhances survival of GC B cells.** The striking enrichment of *GNA13* in all three *BCL6*-based screens prompted us to examine this pathway further. Inactivating mutations of *GNA13* are common in DLBCL and BL[2,25] but rare in other forms of cancer, where, in contrast, amplification may be more common (Supplementary Fig. 3e)[26,27]. Progressive enrichment was seen for eight out of nine gRNAs targeting *GNA13* over different timepoints, with equivalent of greater enrichment to that seen for *TP53* and *CDKN2A* (Fig. 4a). All *GNA13* gRNAs led to effective depletion of GNA13 (Supplementary Fig. 3f), apart from one which was associated with presumed off-target toxicity and further confirmed in a cell line (Supplementary Fig. 3g). GNA13

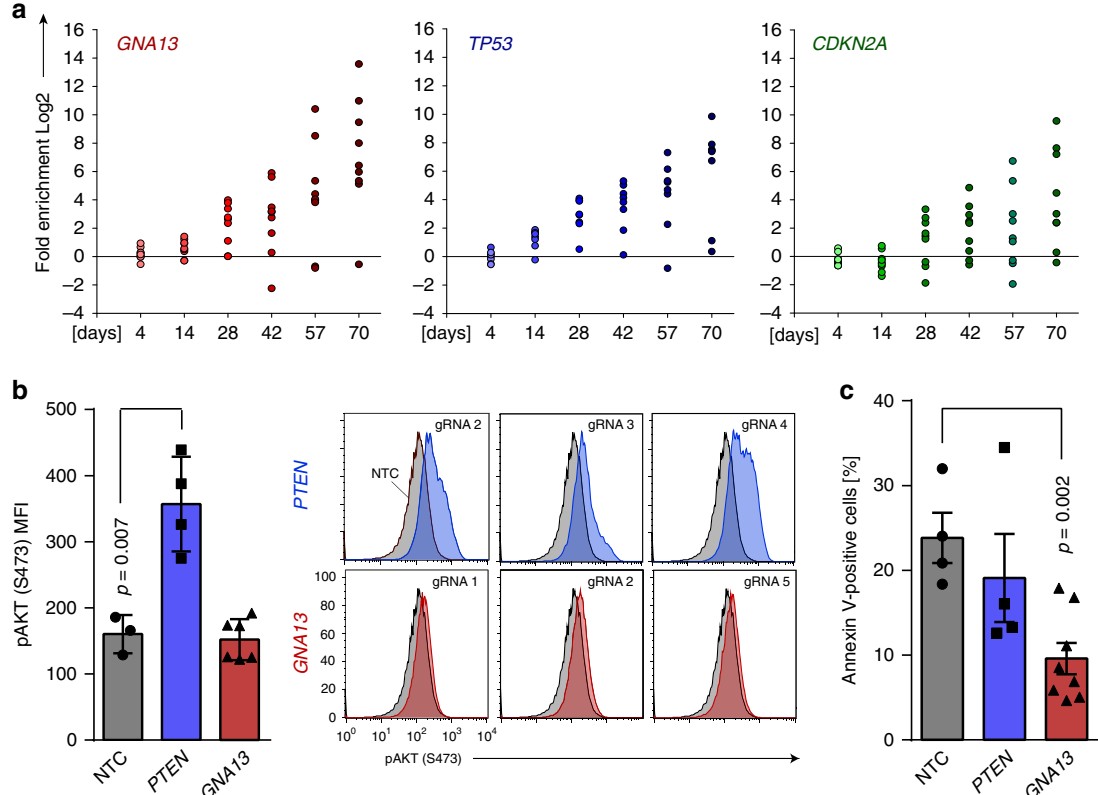

**Fig. 4** *GNA13* revealed as the most potent tumor suppressor gene. **a** Illustrated is log2 fold enrichment for *GNA13*, *TP53*, and *CDKN2A* normalized to non-targeting control gRNAs at the indicated timepoints after transduction with the CRISPR library. Circles represent individual gRNAs for the indicated gene. Data points above the horizontal line are positively enriched. Representative of three experiments in cells from independent donors. Source data are provided as a Source Data file. **b** pAKT levels following *GNA13* and *PTEN* deletion in primary human GC B cells was monitored by intracellular FACS staining for pAKT (S473). A representative example is shown of three *PTEN* gRNAs (blue)/*GNA13* gRNAs (red) against three non-targeting control (NTC) gRNAs (grey). Bar chart illustrates the mean fluorescence intensity of all gRNAs (*GNA13* = 6, *PTEN* = 4, NTC = 3) for the indicated gene (±s.e.m.). The *p* value was calculated from *t*-test. Source data are provided as a Source Data file. **c** Cell survival following *GNA13* and *PTEN* deletion in primary human GC B cells was monitored by annexin-V and 7-aminoactinomycin D (7AAD) staining and analyzed by flow cytometry. Bar chart illustrates annexin V-positive cells of all gRNAs (*GNA13* = 8, *PTEN* = 4, NTC = 4) for the indicated gene (±s.e.m.). The *p* value was calculated by *t*-test. Source data are provided as a Source Data file

acts downstream of the G-protein coupled receptors S1PR2 and P2RY8 and enrichment for both genes was observed in our screens (Fig. 3c). Mouse knockout studies have suggested that suppressed activity of this pathway in lymphoma may allow egress from the GC and increase cell survival secondary to enhanced AKT activity[25,28]. In contrast, other studies suggest a pro-survival effect in DLBCL that is independent of AKT activity[29]. We therefore quantified pAKT levels in ex vivo GC B cells transduced with gRNAs targeting *GNA13*, *PTEN*, or non-targeting controls, and co-cultured on YK6-CD40lg-IL21 feeder cells (Fig. 4b). Although pAKT was increased in *PTEN*-depleted cells, no increase was seen in *GNA13*-depleted cells. However, *GNA13* depletion did lead to a marked reduction in apoptosis in cultured primary GC cells (Fig. 4c), but no change in cell proliferation (Supplementary Fig. 3h). This confirms AKT-independent, enhanced cell survival as the likely explanation for the competitive advantage seen following *GNA13* depletion in this culture system.

**Mutation-directed, in vivo models of human lymphoma**. To examine the ability of the culture-transduction system to recapitulate lymphomagenesis in vivo, we transduced primary, human GC B cells with combinations of oncogenic alterations commonly found in DLBCL (Supplementary Fig. 4a) and injected them in Matrigel into immunodeficient mice (Fig. 5a). Although

sufficient for long-term, feeder-dependent growth in vitro, transduction with *BCL2* and *BCL6*, with or without the addition of a dominant negative *TP53* (P53dd)[30], was insufficient for tumor formation in vivo. However, the addition of a fourth oncogene (*BCL6*, *BCL2*, P53dd, and *CCND3*) led to tumor formation with a median of 112 days (Fig. 5a). The combination of *MYC*, *BCL2*, and P53dd led to tumor formation with a median of 111 days and the combination of *MYC*, *BCL2*, and *BCL6* resulted in tumor formation with a median of 108 days. The most potent combination tested: *MYC*, *BCL2*, P53dd, and *CCND3* resulted in tumor formation in all mice within 38 days. Notably, tumors engrafted with a 100% penetrance and could be derived from multiple donors, excluding the possibility of donor-derived occult mutations contributing to transformation. Flow cytometry showed cells to be strongly positive for markers of all transduced oncogenes, suggesting potent selection during tumorigenesis (Fig. 5b).

Histological examination revealed diffuse sheets of medium to large, atypical lymphoid cells with frequent mitoses, closely mimicking the appearances of human high-grade B cell lymphoma (Fig. 5c, Supplementary Figs. 4b and 5). Immuno-blastic and Burkitt-like appearances were seen in some tumors. Immunohistochemistry showed expression of the B cell markers CD19, CD20, CD79A, and PAX5 in the majority of tumors (Fig. 5c, Supplementary Figs. 4b and 5). In contrast to our

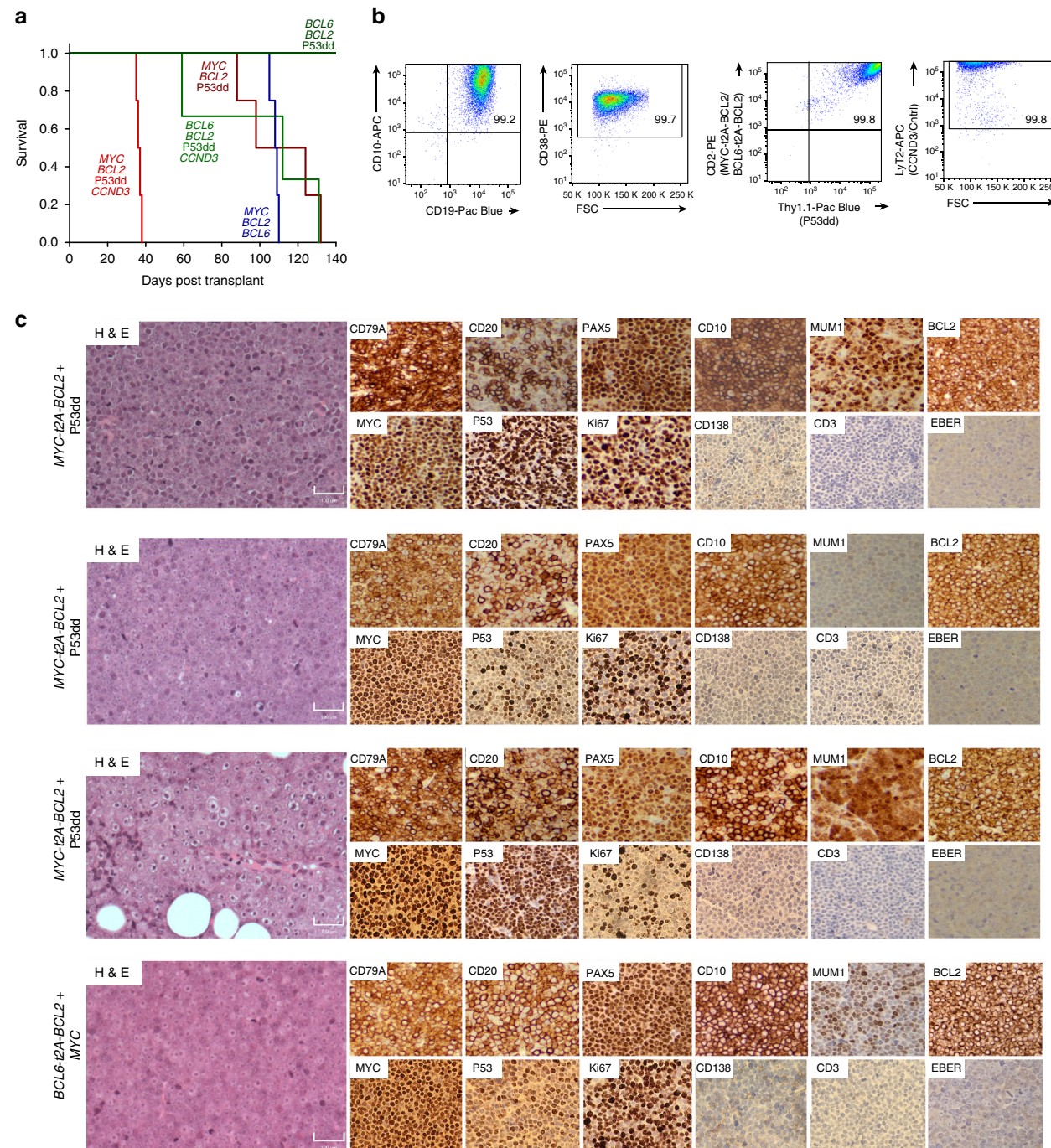

**Fig. 5** Mutation-directed, in vivo models of human lymphoma. **a** Primary human GC B cells (as well as YK6-CD40lg-IL21 cells) transduced with the indicated oncogene cocktails were injected subcutaneously into NOD/SCID/gamma mice (*n* = 3-4 per cohort) and monitored for palpable tumors. Mice were culled when tumors reached 12 mm in size. Survival of the recipient mice is plotted as a Kaplan–Meier curve. Source data are provided as a Source Data file. **b** Cells isolated from tumors were stained for the viral transduction markers CD2 (*MYC-t2A-BCL2* or *BCL6-t2A-BCL2*), Thy1.1 (P53dd), LyT2 (*CCND3*/Cntrl), and B cell markers CD19, CD10, and CD38 and were analyzed by flow cytometry. A representative example is shown. FSC forward scatter. **c** H&E and immunohistochemistry images for the indicated proteins are shown (magnification ×20). Scale bar, 100 μM. Four representative tumors from mice described in **a** are shown

in vitro observations where cells transduced with *MYC* but not *BCL6* downregulated expression of CD20, most *MYC*-driven tumors expressed strong surface CD20. The GC marker CD10 was expressed in approximately half of tumors (Fig. 5c, Supplementary Fig. 4b). Importantly, all tumors were negative for EBER (ISH) confirming that latent EBV genes did not contribute to lymphomagenesis in these tumors. Western blot

and RNA-Sequencing confirmed continued expression of the oncogenic backbone (Supplementary Fig. 6a, b). Harvested tumor cells could be expanded in vitro and serially retransplanted back into immunodeficient mice (Supplementary Fig. 6c), thus functioning as a robust, scalable model system.

To establish the similarity of our synthetic tumors to subsets of bona fide human lymphomas, we sequenced the transcriptome of

16 synthetic tumors generated from primary GC cells transduced with different combinations of oncogenes. We compared their transcriptional profiles to publicly available RNA-Seq data from DLBCL patients enrolled in the GOYA Trial[31], as well as published RNA-Seq from Burkitt lymphoma patients[32]. Principal component analysis was applied to expression of 858 genes[2] that differentiate ABC or GCB DLBCL. While some synthetic tumors clustered closely with ABC DLBCL, the four synthetic tumors where the oncogenic backbone included *BCL6* were the most GCB-like (Fig. 6a). Notably, most synthetic tumors sat in a position intermediate between DLBCL and Burkitt lymphoma, a finding consistent with the forced expression of *MYC*. Indeed, we saw the strong enrichment for two recently described signatures of MYC-driven lymphoma (double hit signature[33] or molecular high-grade signature[34]; Fig. 6b, Supplementary Table 1) suggesting that the particular oncogenic backbone used to create these tumors, generated models with a gene expression profile that closely approximates double hit lymphoma. We observed strong NF-kB activity in the cultured B cells, which we attribute to the CD40 stimulation on feeder cells. However, NF-kB activity was weaker in synthetic tumors, where CD40 stimulation was not present (Supplementary Fig. 7a).

To establish the clonality of tumors, we performed deep sequencing of PCR-amplified immunoglobulin heavy chain variable gene regions to assess the percentage of unique BCR sequences in each sample. This revealed that clonality was increased in primary tumor samples compared to the original donor cells and was also increased in retransplants compared to primary tumors (Fig. 7a). BCR network plots showed that tumors with four oncogenic hits were polyclonal (Fig. 7b). In contrast, cells transduced with just three oncogenic hits, which formed tumors with a longer latency, were clonal (Fig. 7b). This suggests that the combination of four oncogenic events (*MYC, BCL2, CCND3,* and *P53dd*) is by itself sufficient for transformation of human GC B cell. In contrast, further oncogenic events are required for lymphomagenesis in cells transduced with just three of the above constructs. To identify these co-operating oncogenic events, we performed targeted sequencing using a hematological malignancy panel of 292 genes. A subclonal *NRAS* G13A mutation (VAF 0.03) was detected in the oligoclonal tumor arising from *MYC, BCL2,* P53dd transduced cells (Fig. 7c). This mutation became clonal when retransplanted into secondary recipients confirming its role in the pathogenesis of those tumors (Fig. 7c). Mutation at this codon has been reported previously in DLBCL[35] as have other activating mutations of NRAS[2]. We observed copy number increase for the experimentally transduced gene *BCL6* (Supplementary Fig. 7b) but saw no evidence of any significant aneuploidy in any tumor (Supplementary Fig. 7c). In the polyclonal tumors, subclonal mutations with VAF < 0.05 were detected in several genes commonly mutated in DLBCL including a frameshift variant in *S1PR2* and missense mutations in *GNA13, NOTCH2, CREBBP, EP300, SOCS1,* and *BCL6* (Fig. 7c) (Supplementary Data 3). The significance of these mutations to tumor formation is uncertain; however, some of these genes are typical targets of aberrant somatic hypermutation suggesting the possibility of ongoing somatic hypermutation in these lymphomas. To investigate this possibility, we analyzed the variable region sequence of dominant clones detected in the IgH clonality assay. As expected, given their GC origin, almost all clones showed evidence of diversification from the germline V gene sequence (Fig. 7d, e). Importantly, however, clones also showed evidence of ongoing diversification of the hypervariable regions (Supplementary Table 2). This suggests that AID-mediated somatic hypermutation remained active during the process of tumor formation. In addition, analysis of synthetic tumors showed varied expression of the IgH constant region genes

across different tumors, with strong expression in many tumors of IgG and IgA transcripts suggesting that in these tumors, class-switching had occurred before or during tumor development (Supplementary Fig. 7d, e).

Overall, the ability of these tumors to closely recapitulate the appearances of high-grade B cell lymphoma further validates the biological relevance of this system to the study of human lymphoma and provides the opportunity to generate mutation-directed, bespoke in vivo lymphoma models.

## Discussion

The plethora of genomic information generated from next-generation sequencing studies has left us with a need for new experimental systems in which to study the genetics of human lymphoma and to decipher these rich data resources. The availability and suitability of current preclinical models is recognized as a rate-limiting step in translating genomic knowledge into patient benefit[6]. The cell of origin of most aggressive B cell lymphomas, including DLBCL and BL, is the GC B cell[7,8]. We therefore reasoned that non-malignant, human GC B cells should be the input for a system to create genetically defined models of human lymphoma. We describe an optimized system for the culture and transduction of primary, human GC B cells ex vivo. This relies on the provision of microenvironmental survival signals common to that of the GC, as well as the overexpression of combinations of oncogenes common to the pathogenesis of human lymphoma. In particular, this includes *BCL6*, a transcription factor central to the GC reaction as well as an established oncogene in GC-derived lymphoma. A related strategy has been employed previously to expand peripheral blood memory B cells for the purposes of monoclonal antibody engineering[36]. Here, we use genetically altered human, primary, GC B cells for the functional investigation of lymphoma genetics to generate synthetic, in vivo, human models of lymphoma.

A major advantage of using primary GC B cells over established lymphoma cell lines is the ability to investigate defined genetic alterations on a genetically normal background. In particular, this provides a sensitive platform for investigating the ability of specific genetic alterations to increase survival and proliferation. An enhanced oncogenic phenotype is much harder to discern in cell lines where the mutational repertoire is likely to have evolved extensively for optimal in vitro growth. The superior sensitivity of this system, compared to cell lines, to detect alterations associated with increased growth or survival is evidenced by the strong enrichment for TSGs in our CRISPR screen when compared to conventional cell lines. Of the 12 most enriched genes (*TP53, GNA13, CDKN2A, ATRX, NFKBIA, ZFP36L1, ZNF281, PTEN, FBXO11, FUBP1, S1PR2,* and *NFKBIE*), the majority are associated with a tumor suppressor function in lymphoma, established in the literature either from evidence of recurrent genetic inactivation or from their ability to inhibit cancer-promoting pathways[2,7,25,37]. The next 24 most enriched genes included *TET2, TSC1, GSK3B, RB1, CDKN2B, P2RY8,* and *SOCS1,* also implicated as TSGs. Thus, the most enriched genes contained a predominance of recognized TSGs. Although our experiment was not designed for detection of drop-outs, it is notable that the two most depleted genes were those targeting *POU2AF1* and *MYC,* both well-established oncogenes in GC lymphomas[38,39].

Notable absentees from the genes enriching in our CRISPR screens were the histone modifiers *CREBBP, EP300,* and *KMT2D*. These genes show very frequent inactivating mutations in DLBCL and follicular lymphoma[40–43]. These mutations are almost always clonal, suggesting that they arise at an early stage of lymphomagenesis, potentially before the GC stage. Interestingly, mouse

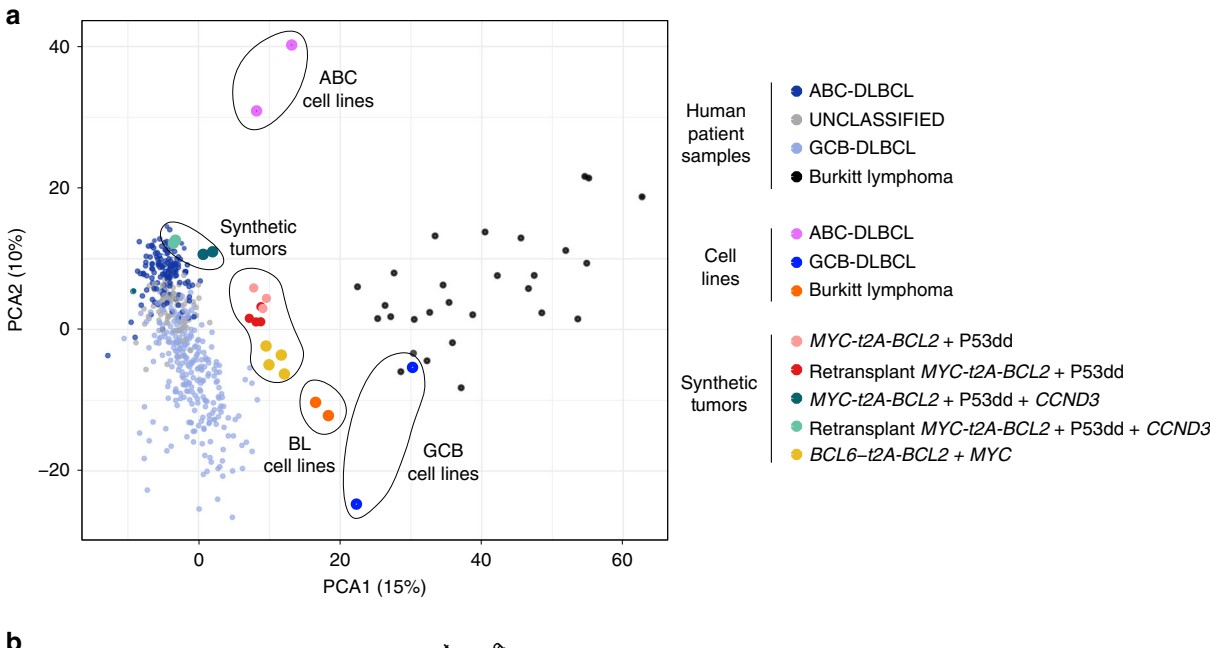

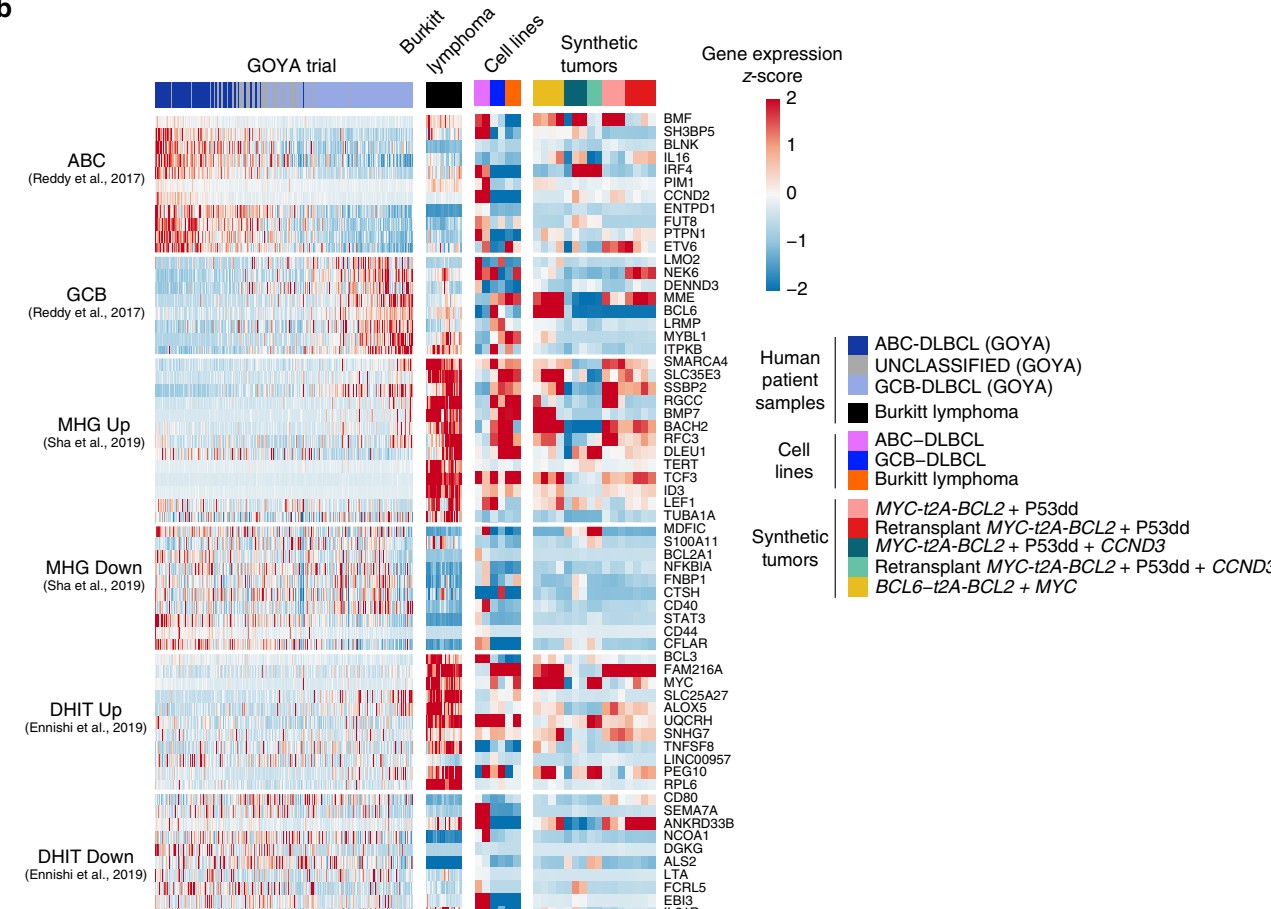

**Fig. 6** Similarity of synthetic tumors to subsets of bona fide human lymphomas and cell lines. **a** PCA plots showing human patient samples (GOYA trial, $n$ = 504; Burkitt lymphoma, $n$ = 28), lymphoma cell lines (Burkitt lymphoma $n$ = 2 (Raji, Mutu), ABC-DLBCL $n$ = 2 (TMD8, HBL1), GCB-DLBCL $n$ = 2 (SUDHL4, DOHH2) and synthetic tumors (BCL6-t2A-BCL2+MYC $n$ = 4, MYC-t2A-BCL2+P53dd $n$ = 3, Retransplant MYC-t2A-BCL2+P53dd $n$ = 4, MYC-t2A-BCL2+P53dd+CCND3 $n$ = 3, Retransplant MYC-t2A-BCL2+P53dd+CCND3 $n$ = 2). PCA was based on expression of 858 genes differentiating between ABC- and GCB-DLBCL subtype as described in Schmitz et al.[2]. **b** Heat map of gene expression of human patient samples (GOYA trial, $n$ = 504, Burkitt lymphoma, $n$ = 28), lymphoma cell lines (Burkitt lymphoma $n$ = 2 (Raji, Mutu), ABC-DLBCL $n$ = 2 (TMD8, HBL1), GCB-DLBCL $n$ = 2 (SUDHL4, DOHH2)) and synthetic tumors (BCL6-t2A-BCL2+MYC $n$ = 4, MYC-t2A-BCL2+P53dd $n$ = 3, Retransplant MYC-t2A-BCL2+P53dd $n$ = 4, MYC-t2A-BCL2+P53dd +CCND3 $n$ = 3, Retransplant MYC-t2A-BCL2+P53dd+CCND3 $n$ = 2). Selected gene signatures are described in Supplementary Table 1. MHG molecular high grade,[34] DHIT double HIT[33]

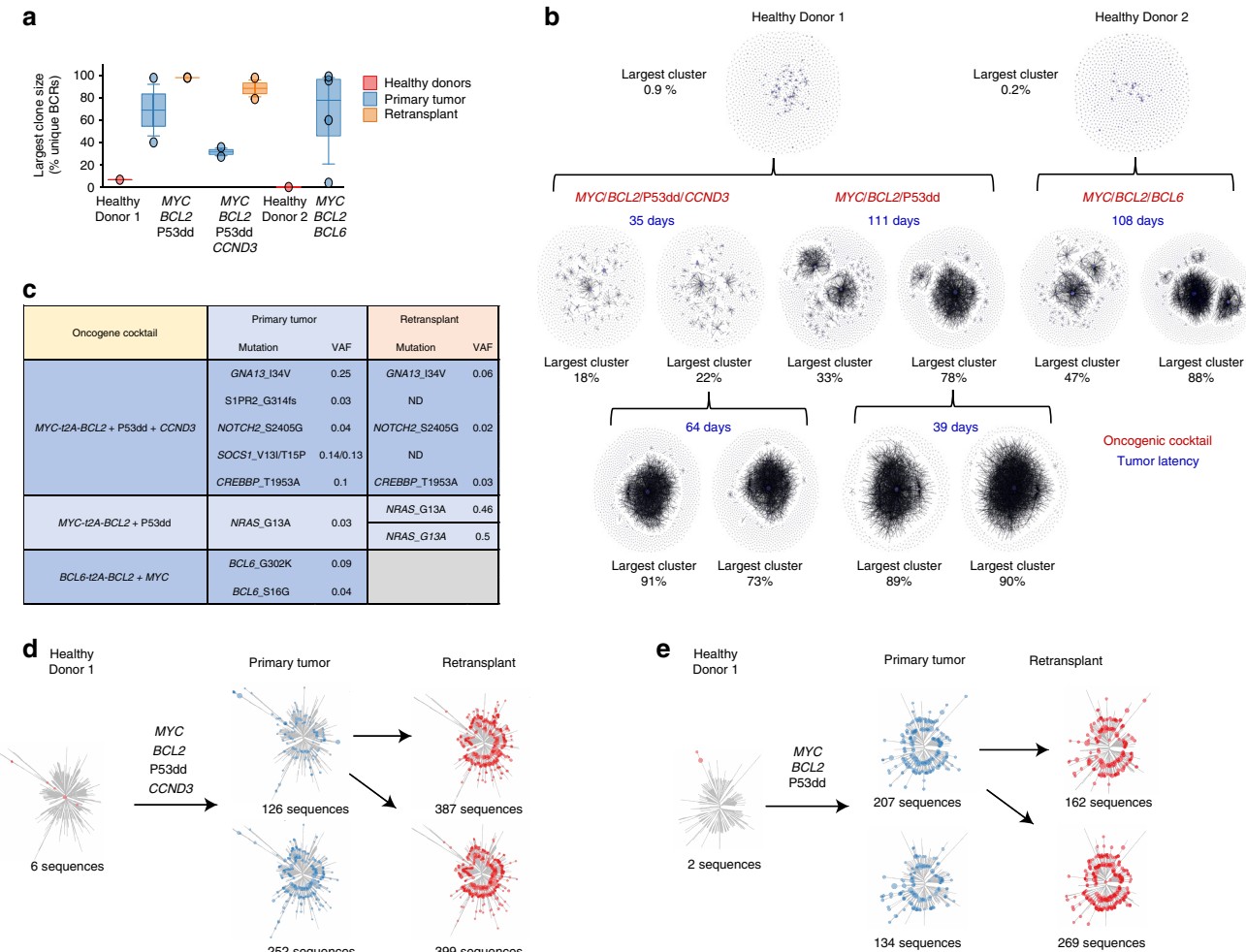

**Fig. 7** Human GC B cells form tumors in immunodeficient mice with ongoing somatic hypermutation. **a** Boxplot shows unique BCR counts (FR1) of primary tumors, retransplant, and healthy donor cells. Source data are provided as a Source Data file. **b** Representative BCR network plots of deep sequenced PCR amplified immunoglobulin variable gene regions from primary tumor samples ($n = 8$) and retransplants ($n = 4$). Each vertex represents a unique sequence, where relative vertex size is proportional to the number of identical reads. Edges join vertices that differ by single-nucleotide non-indel differences and clusters are collections of related, connected vertices. **c** Table shows selected mutations and their relationship in primary and retransplant tumors. **d**, **e** Unrooted phylogenetic trees of representative clonal expansions observed across samples. Each vertex is a unique BCR and branch lengths are estimated by maximum parsimony

models of *Crebbp* and *Kmt2d* deletion show a more pronounced tumor-promoting phenotype when Cre-mediated knockout is induced prior to the GC[44–48]. Therefore, one potential explanation for why these genes do not enrich in our screens is that during lymphomagenesis, the predominant biological effect of these mutations is exerted prior to the GC stage. In contrast, the mutant genes enriched in our screens may reflect those that have the greatest effect in a GC B cell. Interestingly, some of the top hits from our screen show similarity to those associated with transformation of follicular lymphoma into high-grade lymphoma (*GNA13, CDKN2A, TP53, P2RY8, S1PR2*)[42]. We speculate that this might be consistent with our screen detecting those mutations that provide proliferation or survival advantage to an already "corrupted" GC B cell. In developing lymphoma cells, this corruption might reflect pre-GC mutations of *CREBBP*, while in our screen this oncogenic corruption could be provided by the forced expression of *BCL2, BCL6,* or *MYC*.

The most striking finding from our CRISPR screens was the potent enrichment of gRNAs targeting *GNA13*, as well as its upstream receptors *S1PR2* and *P2RY8*. Inactivating mutations of *GNA13* are common in lymphoma, but rarely seen in other forms

of malignancy. Indeed, amplification is more common in solid organ cancers, where *GNA13* is generally considered to act as an oncogene[49]. Thus, its enrichment in our screens reinforces the specificity of this system to the pathogenesis of GC lymphomas. Previous mouse studies have proposed roles for *GNA13* in the migration of GC B cells but reached differing conclusions in its ability to regulate AKT[25,28,29]. Our data reveal an AKT-independent effect in the regulation of survival of human GC B cells, a finding consistent with the greater enrichment of gRNAs targeting *GNA13* over those targeting *PTEN* in our screens.

The enrichment of *GNA13* across experiments performed in *BCL2/BCL6*- transduced GC B cells from three separate human tonsil donors was remarkably consistent. However, when *BCL6* was removed from the oncogenic backbone and replaced with *MYC*, we no longer saw strong enrichment for depletion of *GNA13*. This fits with the distribution of *GNA13* mutations in human DLBCL, which are found predominantly in the EZB subtype described by Schmitz et al., which expresses the highest levels of BCL6[2]. Interestingly, some of the most enriched gRNAs in the context of MYC expression appear to target genes encoding members of the ZFP36 (Tristetraprolin) family of RNA binding

proteins. This finding is consistent with existing biological knowledge of these proteins, which negatively regulate cell cycle[37] and have been demonstrated to oppose cellular transformation in a mouse model of MYC-induced lymphoma[23]. These findings highlight the potential to introduce further changes to the backbone combination in order to study synergy between different sets of cancer genes. We envisage future studies may also remove or replace components of the feeder-based stimulation, for instance to identify factors promoting cytokine-independent growth. The selective pressure imposed could be further altered by the use of pharmacological inhibitors of specific pathways. Future studies might also employ mutant open reading frame (mORF) screens or targeted CRISPR gene editing to introduce specific mutations into endogenous loci.

The relevance of this culture system to the pathogenesis of human lymphoma is underscored most strongly by the ability to recapitulate the appearances of human high-grade B cell lymphoma when cells are engrafted into immunodeficient mice. Notably, our data reveal that the transformation of a human GC B cell appears to require a minimum of four oncogenic hits. The oncogenic backbones used in these experiments employed combinations of *BCL2*, *BCL6*, *MYC*, *TP53*, and *CCND3*, widely accepted as common lymphoma driver genes. RNA-Seq revealed how the oncogenic backbone affected the transcriptional profile, with tumors appearing more GCB-like when *BCL6* was included in the oncogenic backbone. Consistent with the enforced expression of BCL2 and MYC, we saw strong enrichment for signatures of double hit lymphoma[33,34], a subtype of lymphoma characterized by translocation of both *MYC* and *BCL2* and associated with a particularly poor clinical outcome. We anticipate that future studies will further alter the oncogenic backbone to model specific disease subtypes for functional analysis and preclinical drug testing.

The complex genetic heterogeneity of human lymphoma is becoming increasingly evident[2–4]. It is clear that the repertoire of available cell lines does not adequately represent each of the many molecular subtypes predicted from the analysis of sequencing studies. Therefore, the ability to generate mutation-directed tumors in vivo provides an attractive route for patient-personalized preclinical models. A particular advantage over tumor-derived xenograft models is the ability to create paired, syngeneic controls; tumors that are genetically identical other than the presence or absence of a specific mutation. Similar approaches to culture and manipulate human primary cells are proving successful for some solid organ malignancies. However technical limitations have precluded this in B cell lymphoma. We present an extensively optimized, yet inexpensive strategy to employ primary, human, GC B cells for the investigation of lymphoma genomics and to generate bespoke, in vivo models of human lymphoma. This addresses an important bottleneck in translating lymphoma genomic findings into functional understanding that can drive improved patient outcomes and personalized therapy.

## Methods

**Plasmid construction.** The CDS for human gene sequences were cloned into the pBMN-IRES-LyT2 retroviral vector (kind gift of Dr. Louis Staudt, National Cancer Institute, USA) to express *CCND3*, *BCL6*, or *IL21* using synthetic double-stranded DNA from IDT with Gibson Assembly (NEB). For *CCND3*, a mutation in the threonine residue at Threonine 283 (T283A) was included to enhance protein expression[32]. *BCL2, CD40L,* or *MYC* were cloned into MSCV-based vectors using synthetic double-stranded DNA and Gibson Assembly. For overexpression of multiple human genes, CDS sequences were cloned into the MSCV-IRES-huDC2 vector (kind gift of Dr Martin Turner, the Babraham Institute, UK) with the t2A peptides linking genes, such as BCL6-t2A-BCL2, MYC-t2A-BCL2, and P53dd-t2A-BCL2 or an additional p2A peptide when linking three genes such as P53dd-p2A-CDK4_R24C-t2A-BCL2. pBABE-hygro-hTERT was a gift from Bob Weinberg (Addgene plasmid # 1773; http://n2t.net/addgene:1773; RRID:Addgene_1773)[50].

P53dd and CDK4_R24C were cloned into MSCV-IRES-Thy1.1 DEST vector using synthetic double-stranded DNA from IDT with Gibson Assembly. MSCV-IRES-Thy1.1 DEST was a gift from Anjana Rao (Addgene plasmid # 17442; http://n2t.net/addgene:17442; RRID:Addgene_17442)[51].

The MSCV-CAS9-2A-BFP construct was modified from Addgene plasmid #65655 by excision of the IRES/Puro elements and insertion of a 2A-BFP sequence using dsDNA and Gibson assembly. MSCV_Cas9_puro was a gift from Christopher Vakoc (Addgene plasmid # 65655; http://n2t.net/addgene: 65655; RRID:Addgene_65655)[52].

The custom CRISPR library and single gRNAs were cloned into pKLV2-U6gRNA5-Bbsi-PGK-GFP, which was modified from pKLV2-U6gRNA5(Empty)-PGKBFP2AGFP-W. pKLV2-U6gRNA5(Empty)-PGKBFP2AGFP-W was a gift from Kosuke Yusa (Addgene plasmid # 67979; http://n2t.net/addgene: 67979; RRID:Addgene_67979)[53]. Pooled oligos for construction of the lymphoma-focused CRISPR library were obtained from TWIST Bioscience and oligos for single gRNAs were obtained from IDT. To make the GaLV-MuLV fusion envelope constructs, pHIT123[54] (kind gift of Prof Markus Muschen, City of Hope, Los Angeles, CA) containing the retroviral ecotropic envelope, human cytomegalovirus immediate-early promoter and the origin of replication from simian virus 40 was used as the backbone. The viral envelopes GaLV_WT, GaLV_MTR, and GaLV_TR were based on the SEATO strain of GaLV (NP_056791). GaLV_MTR and GaLV_TR contain the 3′ GaLV envelope sequence replaced by the MuLV transmembrane region, cytoplasmic region, and R peptide region and the MuLV cytoplasmic region and R peptide region, respectively[19]. All sequences were purchased from IDT as synthetic double-stranded DNA and inserted by Gibson assembly. All plasmids were verified by capillary sequencing.

**Cell culture.** Cell lines HBL1, BJAB, U2932, TMD8, SUDHL4, DOHH2, Raji, Mutu (all kind gifts from Dr Louis Staudt, National Cancer Institute, USA) and NCIH929 (ATCC CRL-9068) were cultured in Roswell Park Memorial Institute medium (RPMI-1640, Invitrogen, Carlsbad, CA). Primary human GC B cells were cultured in Advanced Roswell Park Memorial Institute medium (Advanced RPMI-1640; Invitrogen, Carlsbad, CA) with GlutaMAX containing 20% FBS, 100 IU/ml penicillin and 100 μg/ml streptomycin and kept at 37 °C in a humidified incubator (5% $CO_2$ and 95% atmosphere).

Lenti-X 293 T Cell Line (Clontech Laboratories, 632180) were cultured in Dulbecco's modified Eagle's medium (DMEM, Invitrogen, Carlsbad, CA) containing 10% FBS, 100 IU/ml penicillin, and 100 μg/ml streptomycin and kept at 37 °C in a humidified incubator (5% $CO_2$ and 95% atmosphere). All cell lines used in this study were confirmed to be free from mycoplasma contamination and identity was verified using a 16-amplicon multiplexed copy number variant fingerprinting assay[24].

**Construction of YK6-CD40Lg-IL21 feeder line.** Discarded human tonsil tissue was obtained after a routine tonsillectomy and handled in accordance with an IRB-approved protocol (2013-0864) at the Asian Medical Center, Seoul, South Korea. The requirement for informed consent was waived by the institutional review board because there was no additional risk to the subjects and all identities were anonymized and completely delinked from unique identifiers. FDC were extracted from tonsils following an established protocol for the creation of HK FDC-like feeder cells[11]. Following mechanical disruption and enzymatic digestion, the released cells were collected and subjected to Ficoll gradient centrifugation for 20 min at 2200 r.p.m. The interface layer that contains FDC was then collected. The cells were resuspended in RPMI 1640 medium and centrifuged at 200 r.p.m. for 10 min at 4 °C over a discontinuous gradient of 7.5% and 3% bovine serum albumin (BSA; A9418; Sigma-Aldrich, St. Louis, MO, US). FDC-enriched fractions were collected from the interface. Cells were washed with HBSS and cultured on tissue culture dishes. Cells isolated and culture after these procedures initially contained large adherent cells with attached lymphocytes. Non-adherent cells were removed and adherent cells replenished with fresh medium every 3–4 days. Adherent cells were trypsinized when confluence was attained. Because of the limited growth in culture, FDC-like cells were immortalized (now termed YK6) through retroviral transduction with pBABE_TERT.Hygro, P53DD_Thy1.1 and CDK4_R24C_Thy1.1. Immortalized YK6 cells were further transduced with hCD40Lg-Puro and IL21-LyT2.

**Purification of human GC B cells.** Fresh, tonsil tissue was sourced from the Addenbrooke's ENT Department, Cambridge with written informed consent of the patient/parent/guardian and processed directly to preserve viability. Ethical approval for the use of human tissue was granted by the Health Research Authority Cambridgeshire Research Ethics Committee (REC no. 07/MRE05/44). GC B cells were purified using the human B cell negative selection isolation Kit II (MACS, Miltenyi Biotec) as per the manufacturer's instructions. The protocol was modified to include negative selection antibodies IgD-BIOT (IADB6; SouthernBiotech 9030-08, 1:100) and CD44-BIOT (F10-44-2; SouthernBiotech 9400-08, 1:100) to remove naïve and memory B cells[55]. Cells were stained for CD38 (HB7; BioLegend #12-0388-42, 1:500), CD20 (2H7; BioLegend #17-0209-41, 1:500), CD19 (HIB19; BioLegend #302223, 1:500), and CD10 (97C5; MiltenyiBiotec, 130-093-450, 1:500) to confirm enrichment of GC B cells. B cell genomic DNA was screened for EBV

status using a quantitative real-time PCR (qPCR) assay[56] and cells from EBV-positive were discarded. qPCR Primer sequences are as follows:

EBV F QP1L 5′-GCCGGTGTGTTCGTATATGG-3′
EBV R QP2L 5′-CAAAACCTCAGCAAATATATGAG-3′

GC B cells were plated onto irradiated YK6-CD40lg-IL21 and split every 2–3 days. Fresh YK6-CD40lg-IL21 cells (irradiated 30 Gy) were added with each split. Primary human GC B cells were cultured in Advanced Roswell Park Memorial Institute medium (Advanced RPMI-1640, Invitrogen, Carlsbad, CA) with GlutaMAX containing 20% FBS, 100 IU/ml penicillin, and 100 μg/ml streptomycin and kept at 37 °C in a humidified incubator (5% $CO_2$ and 95% atmosphere).

**Retroviral and lentiviral production.** Retroviral packaging plasmids pHIT60 (kind gift of Dr. Louis Staudt, National Cancer Institute, USA) and GaLV WT were used as follows: 1 μg pHIT60 (gag-pol), 1 μg GaLV WT (envelope), and 4 μg of a retroviral construct was used to transfect each 10 cm$^2$ dish of HEK-293T, after mixing with 1 ml of Opti-MEM media (Invitrogen) and 18 μl of TransIT-293 (Mirus). For lentivirus transfections, packaging plasmids pCMVDeltaR8.91 and GaLV MTR were used as follows: 8.3 μg pCMVDeltaR8.91 (gag-pol), 2.8 μg GaLV MTR (envelope), and 11 μg of a lentiviral construct per 10 cm$^2$ dish, incubated with 1 ml of Opti-MEM media (Invitrogen) and 33 μl of TransIT-293 (Mirus). For infecting cell lines, pMD2.G (VSV-G envelope) was used instead of GaLV MTR. pMD2.G was a gift from Didier Trono (Addgene plasmid # 12259; http://n2t.net/addgene:12259; RRID:Addgene_12259). The packaging line Lenti-X 293T Cell Line (Clontech Laboratories) was used for all transfections. After 36–48 h, the virus supernatant was filtered through a 0.45 μM filter. If needed, media was replenished and harvested again at 16 h later. For retroviral/lentiviral transduction, 1–2 × 10$^6$ target cells were resuspended with viral supernatant and infected by centrifugation (1500 × $g$, 90 min at 32 °C) with the addition of 10 μg/ml Polybrene (INSIGHT Biotechnology) and 25 μM HEPES (ThermoFisher) in 12- or 24-well plates. Viral supernatant was replaced with fresh media immediately after centrifugation for retroviral infection or after >4 h if transducing lentiviral constructs. Cells were maintained at 37 °C with 5% $CO_2$ for 2–4 days before analyzing by FACS.

**Mouse experiments.** Cultured human GC B cells were injected subcutaneously into the left flank of NSG mice (Jackson laboratories). Up to 10 × 10$^6$ cells were washed and resuspended with Matrigel (Corning) in a 1:1 ratio. Mice were culled when tumors reached 12 mm in size. Tumors were processed immediately after harvest and analyzed by flow cytometry and histology. This research has been regulated under the Animals (Scientific Procedures) Act 1986 Amendment Regulations 2012 following ethical review by the University of Cambridge Animal Welfare and Ethical Review Body (AWERB-PPL number P846C00DB).

**Generation of lymphoma-focused CRISPR guideRNA library.** gRNA sequences were based upon two recent genome-wide libraries[53,57]. Position one of 20 set to G for all gRNAs. Appropriate overlapping sequences (underlined) for Gibson Assembly into the gRNA expression plasmid pKLV2_U6gRNA_Bbsi_PGK_GFP (modified from Addgene #67979) were appended to all 6000 gRNAs. A 70-mer oligo pool was purchased from TWIST BIOSCIENCE as follows:

5′-TATCTTGTGGAAAGGACGAAACACCG-N$_{19}$-GTTTAAGAGCTATGCTGGAAACAGC-3′

N$_{19}$ represents each of the 6000 gRNA sequences. The single-stranded oligo pool was converted to double-stranded DNA by PCR amplification using Q5 Hot Start High-Fidelity 2X Master Mix (NEB) with 3 ng of the oligo pool as a template and primers (Zhang_F and Zhang_R_modified). The following PCR conditions were used as follows: 95 °C for 2 min, 10 cycles of 95 °C for 20 s, 60 °C for 20 s, and 72 °C for 30 s, and the final extension, 72 °C for 3 min. Primer sequences are as follows:

Zhang_F
5′-GTAACTTGAAAGTATTTCGATTTCTTGGCTTTATATATCTTGTGGAAAG GACGAAACACC-3′
Zhang_R_modified
5′-ACTTTTTCAAGTTGATAACGGACTAGCCTTATTTAAACTTGCTATGC
TGTTTCCAGCATAGCTCTTAAAC-3′

The 150 bp PCR product was gel purified from a 2% Agarose gel using the Gel Extraction kit (Qiagen) and eluted in 20 μl EB Buffer. Four Gibson Assembly reactions were performed using 14.4 ng of the purified 150 bp fragment and 200 ng of the BbsI-digested pKLV2_U6gRNA_Bbsi_PGK_GFP with Gibson HiFi DNA Assembly Master Mix (NEB). Gibson Assembly reactions were pooled and column-purified using MinElute PCR purification kit (Qiagen). Eight electroporations were performed using 1 μl of the purified Gibson reaction and 20 μl of Endura Competent Cells (Lucigen). The mixture was transferred to a 0.1 cm cuvette and electroporated at 1.8 kV. Immediately after, 2 ml of prewarmed SOC media was added to each reaction and placed on a shaker at 37 °C for 1 h. The electroporated cells were combined and plated onto 16 24.5 cm$^2$ LB + ampicillin agar plates using ColiRollers Plating Beads (Merck Millipore). Plates were left at 30 °C overnight and plasmid DNA was purified using a Plasmid Maxi kit (Qiagen).

**Transduction of CRISPR library and generation of gRNA sequencing libraries.** GC B cells were transduced with the backbone oncogene cocktail and Cas9-BFP retrovirus until Cas9-BFP reached between 50% and 80%. The number of cells transduced with gRNA library was adjusted to take account of the percentage of Cas9-expressing cells and target MOI of 0.3 in order to maintain representation of >1000× the size of the library. Four days after transduction, BFP and GFP expression was analyzed by flow cytometry and at each harvest timepoint going forward. A minimum of 1000× representation was maintained at each passaging step. Cells were harvested every 14 days. Genomic DNA extraction was conducted as described previously[58] which is as follows: 600 μl Lysis Buffer and 15 μl Proteinase K were used to re-suspend cells and left at 65 °C for 15 min. Depending on cell pellet, Lysis Buffer and Proteinase K volume was scaled up and incubated until pellet completely lysed. Isopropanol was used to precipitate genomic DNA and further re-suspended in TE Buffer. Illumina sequencing was performed as follows[53,59]. For sequencing of all gRNAs in the CRISPR library, primers (gLibrary-HiSeq_50bp-SE-U1 and −L1) were used to amplify the region containing the gRNA. Primer sequences are as follows:

gLibrary-HiSeq_50bp-SE-U1
5′-ACACTCTTTCCCTACACGACGCTCTTCCGATCTCTTGTGGAAAGGACGAAACA-3′
gLibrary-HiSeq_50bp-SE-L1
5′-TCGGCATTCCTGCTGAACCGCTCTTCCGATCTCTAAAGCGCATGCTCCAGAC-3′

For this PCR, it is crucial to use sufficient genomic DNA to capture every gRNA in the cell population. This depends on the complexity of populations to be analyzed. For human cells, 10$^6$ cells contain around 6.6 μg of genomic DNA (assuming normal copy number). Therefore, the number of cells (millions) harvested × 6.6 μg will correspond to the amount of genomic DNA needed in the first PCR. For example, if a cell population was 30% double positive for CAS9-BFP and CRISPR library-GFP, then 20 × 10$^6$ cells were harvested to achieve 1000× coverage (6000 guides × 1000 coverage/0.3). In this case 131 μg (20 × 6.6) of genomic DNA was used in the first PCR, with a maximum of 10 μg per 50 μl reaction. Therefore, 13 independent PCR reactions were performed using 10 μg of genomic DNA per reaction with Q5 Hot Start High-Fidelity 2x Master Mix. The following PCR conditions were used: 98 °C for 30 s, 20–24 cycles of 95 °C for 10 s, 61 °C for 15 s, and 72 °C for 20 s, and the final extension, 72 °C for 2 min. Five microliters from each PCR reaction were run on 2% Agarose gel and PCR was run for a few more cycles if there was no PCR product or PCR bands were still faint. Next, 10 μl from each individual PCR reaction per sample was taken, pooled and purified using QIAquick PCR Purification Kit (Qiagen). DNA was eluted in 50 μl EB buffer (Qiagen) and concentration was quantified on the nanodrop. In the second PCR, nextgen sequencing adaptors (P5, P7) compatible with Illumina's HiSeq4000 and a barcode were added. One nanogram of the purified PCR product was used with NEBNext Q5 Hot Start HiFi PCR Master Mix with the following conditions: 98 °C for 30 s, 9–12 cycles of 98 °C for 10 s, 65 °C for 75 s and the final extension, 65 °C for 5 min. Forward primer named Indexing Adapter PE 1.0 and different reverse indexing primers (iPCRtagT1-56) were used in this second PCR. A different reverse indexing primer was used for each sample. Primer sequences are as follows:

Indexing Adapter PE 1.0
5′-AATGATACGGCGACCACCGAGATCTACACTCTTTCCCTACACGACGCTCTTCCGATC*T-3′
iPCRtagT1
5′-CAAGCAGAAGACGGCATACGAGATAACGTGATGAGATCGGTCTCGGCATTCC
TGCTGAACCGCTCTTCCGATC*T-3′
iPCRtagT2
5′-CAAGCAGAAGACGGCATACGAGATAAACATCGGAGATCGGTCTCGGC
ATTCCTGCTGAACCGCTCTTCCGATC*T-3′
iPCRtagT3
5′-CAAGCAGAAGACGGCATACGAGATATGCCTAAGAGATCGGTCTCGGCATTCCTGCTGAACCGCTCTTCCGATC*T-3′
iPCRtagT4
5′-CAAGCAGAAGACGGCATACGAGATAGTGGTCAGAGATCGGTCTCGGCATTCCTGCTGAACCGCTCTTCCGATC*T-3′
iPCRtagT5
5′-CAAGCAGAAGACGGCATACGAGATACCACTGTGAGATCGGTCTCGGCATTCCTGCTGAACCGCTCTTCCGATC*T-3′
iPCRtagT6
5′-CAAGCAGAAGACGGCATACGAGATACATTGGCGAGATCGGTCTCGGCATTCCTGCTGAACCGCTCTTCCGATC*T-3′
iPCRtagT7
5′-CAAGCAGAAGACGGCATACGAGATCAGATCTGGAGATCGGTCTCGGCATTCCTGCTGAACCGCTCTTCCGATC*T-3′
iPCRtagT8
5′-CAAGCAGAAGACGGCATACGAGATCATCAAGTGAGATCGGTCTCGGCATTCCTGCTGAACCGCTCTTCCGATC*T-3′
iPCRtagT9
5′-CAAGCAGAAGACGGCATACGAGATCGCTGATCGAGATCGGTCTCGGCATTCCTGCTGAACCGCTCTTCCGATC*T-3′

iPCRtagT10
5′-CAAGCAGAAGACGGCATACGAGATACAAGCTAGAGATCGGT
CTCGGCATTCCTGCTGAACCGCTCTTCCGATC*T-3′
iPCRtagT11
5′-CAAGCAGAAGACGGCATACGAGATCTGTAGCCGAGAT
CGGTCTCGGCATTCCTGCTGAACCGCTCTTCCGATC*T-3′
iPCRtagT12
5′-CAAGCAGAAGACGGCATACGAGATAGTACAAGGAG
ATCGGTCTCGGCATTCCTGCTGAACCGCTCTTCCGATC*T-3′
iPCRtagT13
5′-CAAGCAGAAGACGGCATACGAGATAACAACCAGAGA
TCGGTCTCGGCATTCCTGCTGAACCGCTCTTCCGATC*T-3′
iPCRtagT14
5′-CAAGCAGAAGACGGCATACGAGATAACCGAGAGAGATCGGT
CTCGGCATTCCTGCTGAACCGCTCTTCCGATC*T-3′
iPCRtagT15
5′-CAAGCAGAAGACGGCATACGAGATAACGCTTAGAGATCGGTCT
CGGCATTCCTGCTGAACCGCTCTTCCGATC*T-3′
iPCRtagT16
5′-CAAGCAGAAGACGGCATACGAGATAAGACGGAGAGATCGGTCTC
GGCATTCCTGCTGAACCGCTCTTCCGATC*T-3′
iPCRtagT17
5′-CAAGCAGAAGACGGCATACGAGATAAGGTACAGAGATCGGTCTC
GGCATTCCTGCTGAACCGCTCTTCCGATC*T-3′
iPCRtagT18
5′-CAAGCAGAAGACGGCATACGAGATACACAGAAGAGATCGGTCTCG
GCATTCCTGCTGAACCGCTCTTCCGATC*T-3′
iPCRtagT19
5′-CAAGCAGAAGACGGCATACGAGATACAGCAGAGAGATCGGTCTC
GGCATTCCTGCTGAACCGCTCTTCCGATC*T-3′
iPCRtagT20
5′-CAAGCAGAAGACGGCATACGAGATACCTCCAAGAGATCGGTCT
CGG
CATTCCTGCTGAACCGCTCTTCCGATC*T-3′
iPCRtagT21
5′-CAAGCAGAAGACGGCATACGAGATACGCTCGAGAGATCGGTCT
CGG
CATTCCTGCTGAACCGCTCTTCCGATC*T-3′
iPCRtagT22
5′-CAAGCAGAAGACGGCATACGAGATACGTATCAGAGATCGGTCTC
GGCATTCCTGCTGAACCGCTCTTCCGATC*T-3′
iPCRtagT23
5′-CAAGCAGAAGACGGCATACGAGATACTATGCAGAGATCGGTCT
CGGCATTCCTGCTGAACCGCTCTTCCGATC*T-3′
iPCRtagT24
5′-CAAGCAGAAGACGGCATACGAGATAGAGTCAAGAGATCGGTCTC
GGCATTCCTGCTGAACCGCTCTTCCGATC*T-3′
iPCRtagT25
5′-CAAGCAGAAGACGGCATACGAGATAGATCGCAGAGATCGGTCT
CGGCATTCCTGCTGAACCGCTCTTCCGATC*T-3′
iPCRtagT26
5′-CAAGCAGAAGACGGCATACGAGATAGCAGGAAGAGATCGGTCTC
GGCATTCCTGCTGAACCGCTCTTCCGATC*T-3′
iPCRtagT27
5′-CAAGCAGAAGACGGCATACGAGATAGTCACTAGAGATCGGTCT
CGG
CATTCCTGCTGAACCGCTCTTCCGATC*T-3′
iPCRtagT28
5′-CAAGCAGAAGACGGCATACGAGATATCCTGTAGAGATCGGTCT
CGG
CATTCCTGCTGAACCGCTCTTCCGATC*T-3′
iPCRtagT29
5′-CAAGCAGAAGACGGCATACGAGATATTGAGGAGAGATCGGTCTCG
GCA
TTCCTGCTGAACCGCTCTTCCGATC*T-3′
iPCRtagT56
5′-CAAGCAGAAGACGGCATACGAGATGTACGCAAGAGATCGGTCTC
GGC
ATTCCTGCTGAACCGCTCTTCCGATC*T-3′

Five microliters from each PCR reaction were run on 2% agarose gel and checked for visible PCR bands. The PCR products were purified with Agencourt AMPure XP beads in a PCR-product-to-bead ratio of 1:0.7 and eluted in 30 μl EB Buffer (Qiagen). The purified libraries were quantified, pooled, and sequenced on Illumina HiSeq4000 by 50-bp single-end sequencing. Two custom sequencing primers were used here: iPCRtagseq which reads through the indices and U6-Illumina-seq2 which reads through the gRNA sequence. Enriched gRNAs were defined based upon enrichment relative to the plasmid pool counts. Purified libraries were quantified, pooled, and sequenced on Illumina HiSeq4000 by 50-bp single-end sequencing with the following primers:

iPCRtagseq 5′-AAGAGCGGTTCAGCAGGAATGCCGAGACCGATCTC-3′
U6-Illumina-seq2 5′-TCTTCCGATCTCTTGTGGAAAGGACGAAACACCG-3′

**Computational analysis of CRISPR screens**. Raw reads were normalized to a total number of reads in a sample as follows:

$$F_{igtr} = \frac{N_{igtr}}{\sum_i N_{igtr}} \tag{1}$$

denotes the raw sequencing reads of gRNA $i$ of gene $g$ at time $t$ in replicate $r$. For each gRNA the $Z$-score of $\log_2$ fold change between plasmid library and late sample, $Z_{igr}$ is given by

$$\begin{aligned}\text{sgRNA}_{\Delta igr} &= \log_2 \frac{F_{igr\,L}}{F_{igr\,P}}, \\ Z_{igr} &= \frac{\text{sgRNA}_{\Delta igr} - \overline{\text{sgRNA}_{igr}}}{\sigma_{\text{sgRNA}_{\Delta gr}}},\end{aligned} \tag{2}$$

Finally, CRISPR score$_g$, which represents the magnitude and direction of a fitness of a gene $g$ between the two time points is

$$\text{CRISPR score}_g = \frac{1}{RL_g} \sum_{r=1}^{R} Z_{igr}, \tag{3}$$

where $L_g$ denotes the number of sgRNA of gene $g$ in replicate $r$ and $R$ is the number of available replicates.

**RNA-sequencing**. Total RNA from cells was extracted using NucleoSPIN RNA from Macherey-Nagel and cDNA was produced from 500 ng of total RNA using qScript$^{TM}$ cDNA SuperMix (Quanta Biosciences). RNA-seq library was prepared using the NEBNext Poly (A) mRNA Magnetic Isolation Module (E7490) and NEBNext Ultra Directional RNA Library Prep Kit for Illumina (E7420) according to the manufacturer's instructions. NEBNext Multiplex Oligos for Illumina (E7500) was used for indexing and sequenced on a HiSeq4000 by 50-bp single-end sequencing. RNA-Seq were mapped to the hg38/GRCh38 reference human genome using splice-aware aligner STAR 2.5.3a[60] in two pass-mode. The genome index was built with GENCODE v.28 comprehensive gene annotation set. Uniquely mapped reads were assigned to genes with RSubread package[61] allowing for assignment of a read to more than one overlapping features. At least 25 of overlapped bases were required to assign a read to a gene. Genes with low counts were filtered out with a threshold of minimum 128 counts in at least 25% of samples. Gene expression measurements were TPM normalized. To remove effect of technical variation between different RNA-Seq runs, variance stabilizing transformation was applied as implemented in the DESeq2 (ref. [62]) package.

**Barcoded overexpression experiments**. The CDS for human gene sequences (BCL6 WT, BCL6_G559R, BCL6_H641R, BCL6_R585W, BCL6_P586A, IRF8 WT, IRF8_N87Y, IRF8_T80A, IRF8_380_stop, IRF8_S55A, MEF2B WT, MEF2B_D83V, MEF2b Y69H) were cloned into barcoded pBMN-IRES-LyT2 retroviral vector using NEBuilder® HiFi DNA Assembly. Primary human GC B cells were retrovirally transduced with BCL2, followed by infection with barcoded overexpression genes and pooled 4 days after transduction, and then grown in competitive culture. Genomic DNA was collected at day 4 and approximately every 14 days after. Genomic DNA extraction was conducted as described in the Methods section "Transduction of CRISPR library and generation of gRNA sequencing libraries". Individual indices (Truseq Small RNA Index Sequences) were added using Q5 Hot Start High-Fidelity 2x Master Mix. The purified library was quantified, pooled and sequenced on Illumina MiSeq by 50-bp single-end sequencing.

**Computational analysis of barcoded overexpression experiments**. Relative abundances $F_{ictr}$ of a construct in a pooled competitive culture were computed as follows:

$$F_{ictr} = \frac{N_{ictr}}{\sum_i N_{ictr}}, \tag{4}$$

where $N_{ictr}$ denotes the raw sequencing counts of clone $i$ of constructs $c$ at time $t$ in replicate $r$. The average relative abundance of construct $c$ at time $t$ is given by

$$M_{ct} = \frac{1}{RL_{ct}} \sum_{r=1}^{R} F_{ictr}, \tag{5}$$

where $L_{ct}$ denotes the number of clones of construct $c$ at time $t$ and $R$ is the number of available replicates.

**BCR amplification**. PCR amplification of DNA from synthetic lymphoma tumors (100 ng input) was performed with 1 μl of JH reverse primer (10 μM) and 1 μl of FR1 forward primer set pools (10 μM per primer, provided by Sigma Aldrich) using 25 μl of Q5 Hot Start Master Mix (2X) (New England Biolabs) for a total volume reaction of 50 μl. The following PCR program was used: 5 min at 95 °C; three cycles of 5 s at 98 °C and 2 min at 72 °C; three cycles of 5 s at 98 °C, 10 s at 65 °C, and 2 min at 72 °C; and 25 cycles of 20 s at 98 °C, 30 s at 60 °C, and 2 min at 72 °C; with a final extension cycle of 7 min at 72 °C on a Mastercycler nexus Thermocycler (Eppendorf) (modified from ref. [63]). MiSeq libraries were generated using KAPA Hyper Prep Kit (KAPA Biosystems) incorporating KAPA Dual-Indexed Adapter

for Illumina MiSeq platforms following the manufacturer instructions. MiSeq reads were filtered for base quality (median Phred score > 32) using the QUASR program (http://sourceforge.net/projects/quaasi/) and for length (300 bp paired-end)[64]. The computational pipeline MRD Assessment and Retrieval Code in Python (MRDARCY) was then used to analyze BCRs, followed by secondary rearrangement analysis in which the relative frequencies of each IgHV gene were determined by BLAST using the ImMunoGeneTics (IMGT) reference gene database. The following primers were used:

JH reverse 5′-CTTACCTGAGGAGACGGTGACC-3′
VH1-FR1 forward 5′-GGCCTCAGTGAAGGTCTCCTGCAAG-3′
VH2-FR1 forward 5′-GTCTGGTCCTACGCTGGTGAAACCC-3′
VH3-FR1 forward 5′-CTGGGGGGTCCCTGAGACTCTCCTG-3′
VH4-FR1 forward 5′-CTTCGGAGACCCTGTCCCTCACCTG-3′
VH5-FR1 forward 5′-CGGGGAGTCTCTGAACATCTCCTGT-3′
VH6-FR1 forward 5′-TCGCAGACCCTCTCACTCACCTGTG-3′

**High-throughput sequencing and analysis of heavy chain immunoglobulin**. Deep sequencing of PCR amplified immunoglobulin heavy chain variable gene regions and BCR network generation algorithm and network properties were performed as follows[63]. Each vertex represents a unique sequence, where relative vertex size is proportional to the number of identical reads. Edges join vertices that differ by single-nucleotide non-indel differences and clusters are collections of related, connected vertices. Ig gene usages and sequence annotation were performed in IMGT V-QUEST, where repertoire differences were performed by custom scripts in Python.

For the visual representations of the BCR repertoires, BCR network subsampling was performed using the cluster-enforced linkage sampling (CC) method to preserve the overall clonal structure. Briefly, the CC algorithm employs three steps to account for loss of connectivity between vertices in clusters during sampling:

*Vertex selection*: Vertices were reselected until the number of desired clusters in the original network G are represented.

*Cluster-vertex migration*: For each cluster in the original network which contains more than one vertex that was sampled, vertices were reselected such that the cluster connectivity is retained in the sampled network.

*Induced graph formation*: Graph induction selects the set of edges (Es) to be included in the sampled graph. Total graph induction is used in CC, selecting all edges incident on the sampled vertices are included in the sampled graph.

This process was repeated 20 times, and the subsample that most closely represented the true (unsampled) maximum cluster size was retained and plotted.

IGHV gene editing analyses were performed in a similar manner to. For all BCRs, stem regions were identified (defined as N-IgHD-N-IgHJ regions starting 3 bp downstream of the IgHV gene boundary). The number of unique BCR sequences sharing stem regions but with different IgHV gene usage (>95% difference in sequence identity in the IgHV region) and with different 5′ of the junctional region (defined as IgHV(last 3pb)-N-IgHD-N-IgHJ) was determined and compared to the total number of unique BCRs to give the percentage *IgHV* replacement. Sequences with joining regions (N-IgHD-N-IgHJ regions) shorter than eight nucleotides were excluded from this percentage due to potential of germline encoded receptors.

**Mutation analysis**. To identify somatic mutations across synthetic lymphoma tumors a hybrid-capture platform was used with a bait set[58] (SureSelect, Agilent, UK, ELID # 0731661) of 292 genes frequently mutated in hematological malignancies. After hybridization-based sequence enrichment (SureSelect$^{HSXT}$, Agilent), high-throughput sequencing was performed on the Illumina HiSeq 4000 platform.

**Sequencing data alignment**. DNA sequencing reads were aligned to the GRCh37d5 according to the workflow described at Samtools webpage (http://www.htslib.org/workflow/) as follows: For mapping the data to a given reference genome BWA-MEM[65] 0.7.17, followed by Samtools[66] 1.9 for cleaning up read pairing information and flags on SAM records. For improvement of the mapped data Broad's GATK[67] Realigner 3.8.1 was used in order to reduce the number of miscalls of INDELs, followed by Picard 2.18.25 (http://broadinstitute.github.io/picard) for identifying duplicates.

**Variant calling for substitutions and indels**. Single base substitutions and short insertions and deletions were called using GATK[67] 4.1 Mutect2 based on the tutorials available at Broad Institute website (https://gatkforums.broadinstitute.org/gatk/discussion/11136/how-to-call-somatic-mutations-using-gatk4-mutect2). The mutant variants were annotated using Variant Effect Predictor[68] from ENSEMBL version 95.

**Copy number analysis**. Copy number analysis was performed using GeneCN (https://github.com/wwcrc/geneCN).

**FACS (fluorescence-activated cell sorting)**. Cells were stained with fluorophore-labeled antibodies in 2% BSA in PBS according to the manufacturer's instructions.

The stained/or unstained cells were analyzed on the LSRII (BD). For cell counting, CountBright Absolute Counting Beads (ThermoFisher) were used according to the manufacturer's instructions and analyzed on the LSRII (BD). For dead cell apoptosis analysis, APC-conjugated Annexin V/Dead Cell Apoptosis Kit (BioLegend 640930) was used for the detection of apoptotic cells according to the manufacturer's instructions. Externalization of phosphatidylserine (Annexin V, APC Conjugate; BioLegend, 1:20) and DNA content (7-AAD; BioLegend,1:20) were measured and gating on all cells was used for further analysis. Cell cycle analysis was performed using the Vybrant® DyeCycle™ Ruby Stain (ThermoFisher V10309, 1:500, Final stain concentration 5 μM) according to the manufacturer's instructions. Cells were treated with Nocodazole (1 μg/ml) 24 h prior to staining. Stained cells were analyzed by gating on cells in the G2 phase using FlowJo software.

Intracellular staining of phosphorylated AKT was performed as follows: Cell suspension and pre-warmed Fixation Buffer (BD Cytofix) was gently mixed in a 1:1 ratio and incubated at 37 ºC for 15 min. Cell suspension was pelleted and washed with PBS twice at 350*g* for 5 min. Ice-cold True-Phos perm buffer (BD Cytofix) was added dropwise to the cell pellet while vortexing, followed by incubation at −20 ºC for at least 60 min. Cells were further washed twice and resuspended in FACS buffer (PBS + 2% FBS) containing the appropriate antibody at a dilution of 1:50 (Phospho-Akt Ser473, Cell Signaling, #11962). After staining for 30 min, cells were washed and resuspended in FACS buffer followed by analysis on the LSRII (BD).

The following antibodies were used: CD38 (HB7, #12-0388-42, 1:500), CD20 (2H7, #17-0209-41, 1:500), CD19 (HIB19, #302223/302211/302205, 1:500), CD10 (97C5, MiltenyiBiotec, 130-093-450, 1:500), CD2 (RPA-2.10, #300207/300235, 1:500), CD90.1 Thy1.1 (OX-7, #202529, 1:500), CD154 (24-31, #310805, 1:500), CD8a (53-6.7, #100712, 1:500), CD22 (HIB22, #302510, 1:500), CD80 (2D10, #305219, 1:500), CD95 (DX2, #305629, 1:500), CD86 (IT2.2, #305419, 1:500), CXCR4 (12G5, #306505, 1:500), IgM (MHM-88, #314506, 1:500) and IgG (#H10164, 1:500). All antibodies were purchased from BioLegend if not otherwise stated. Gating strategies can be found in Supplementary Fig. 8.

**Western blotting**. Western blotting was performed as follows:[58] CelLytic buffer (Sigma) supplemented with 1% protease inhibitor cocktail (Roche), 1% phosphatase inhibitor cocktail (Calbiochem Millipore), and 1 mM PMSF (Life Technologies) was used to lyse cells followed by loading the same amount of protein per sample on NuPAGE (Invitrogen) 4–12% Bis-Tris gradient gels and were further transferred on PVDF membranes (Invitrogen). Primary antibodies were used with the HRP immunodetection system (Life Technologies/Millipore). The following antibodies were used: Beta-actin (13E5; Cell Signaling Technology #4970, 1:10000), GNA13 (EPR5436; Abcam ab128900, 1:1000), BCL-6 (D65C10; Cell Signaling Technology #5650, 1:500), c-MYC (N-262; Santa Cruz sc-764, 1:500), and BCL-2 (7/Bcl-2; Becton Dickinson Biosciences 610539, 1:500). Uncropped western blots can be found in the Source Data file.

**Patient samples**. Gene expression data (RNA-Seq) from DLBCL and Burkitt lymphoma patients were downloaded from Gene Expression Omnibus (GEO), accession number: GSE125966 (ref. [31]) and from Sequence Read Archive (SRA), accession number: SRA048058 (ref. [32]). Datasets were TMM normalized and adjusted for batch effect with ComBat function from sva package (3.30.1)[69].

**Reporting summary**. Further information on research design is available in the Nature Research Reporting Summary linked to this article.

# Data availability
Gene expression data has been uploaded to the EGA database under the accession number EGAS00001003560. Deep sequencing of PCR amplified immunoglobulin heavy chain variable gene regions has been submitted to the SRA under the BioProject ID: PRJNA551148 [https://www.ncbi.nlm.nih.gov/Traces/study/?acc=PRJNA551148]. All the other data supporting the findings of this study are available within the article and its supplementary information files and from the corresponding author upon reasonable request. A reporting summary for this article is available as a Supplementary Information file.

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

## Acknowledgements

D.H. was supported by a Clinician Scientist Fellowship from the Medical Research Council (MR/M008584/1). Research in the Hodson laboratory is supported by the Kay Kendall Leukaemia Fund, The Addenbrooke's Charitable Trust, Cancer Research UK, and a research scholarship from Gilead Sciences. The Hodson laboratory receives core funding from Wellcome and MRC to the Wellcome-MRC Cambridge Stem Cell Institute. We thank Alice Mitchell and the ENT Department at Addenbrooke's Hospital, Cambridge for their assistance in the collection of primary tonsil tissue. We are grateful to Joanna Baxter and Cambridge Blood and Stem Cell Bank for collection and storage of primary tonsils samples and to the staff of the Central Biomedical Services for animal housing and care. We thank Craig McDonald for technical assistance with video microscopy, Mairi Shepherd for graphical illustrations and Martin Dyer, Ingo Ringshausen, and Reuben Tooze for critical reading of the manuscript.

## Author contributions

R.C. designed and performed the experiments and analyzed data. M.D.R., J.G., M.L.C., R.F., Z.U., H.R., and A.M. performed experiments. J.K. analyzed RNA-seq experiments. J.M.L.D. analyzed CRISPR screen experiments. S.L.C. analyzed copy number alterations. R.J.M.B. analyzed whole genome sequencing and clonality experiments. H.E. provided immunohistochemistry support. H.K.P. and C.S.P. provided YK-feeder cells. P.A.B., G.V. and B.J.P.H. provided conceptual input to the study. D.J.H. designed the experiments, provided clinical expertise, directed the research, and wrote the manuscript with contributions from R.C.

## Competing interests

D.J.H.: Research funding from Gilead Sciences; Consultancy for Karus Therapeutics. P.A.B.: consultancy for Karus Therapeutics (Oxford, UK), OncoDNA (Gosselies, Belgium), and Everything Genetic (London, UK). The remaining authors declare no competing interests.
