## [Peer Review File · Nature Communications]

Reviewers' comments:

Reviewer #1 (Remarks to the Author):

This manuscript by Caeser and colleagues reports the generation of an innovative model system to grow and expand human tonsillar GC B lymphocytes in the FDC-like feeder line HK genetically modified to express Cd40lg and to secrete IL21. Remarkably, transduction of these primary cells with two different combinations of lymphoma-associated oncogenes (MYC+BCL2 or BCL6+BCL2) led to in vitro malignant transformation. These tumors recapitulated phenotypical and gene expression features of human GC-derived lymphomas. High-throughput CRISPR-based screening using gRNAs targeting 692 genes mutated or deleted in lymphoma revealed well-known TSGs (P53, PTEN, CDK2A) and the GNA13 gene, also commonly mutated in GC-derived DLBCL (and of other downstream proteins S1PR2 and P2RY8), which collectively confirms the validity of the system to investigate GC lymphomagenesis. Transduced cells were capable to grow in immunodeficient mice, and retained the principal features of GCB and ABC DLBCLs.

Overall, this is an interesting, outstanding technical manuscript that reports for the first time (as far as I know) the oncogenic transformation of human primary GC B lymphocytes ex vivo upon transduction of known oncogenic drivers. This extraordinary, reliable, simple setting will allow the scientific community not only to continue investigating GC-derived lymphoma biology and development, but also testing drug combinations pre-clinically.

A number of doubts, observations and suggestions follow:

- 1) Regarding DLBCLs developed in vitro and in mice, were these classified as ABC or GCB subtypes and were these subtypes correlated with specific mutation combinations?
- 2) While signs of IGVH somatic hypermutation are shown in the tumors, did class-switch recombination occur? Did tumor cells express surface IgM? Did the authors stain for IgG expression? This could be clinically relevant, as half of DLBCLs in patients show class-switched phenotypes, but most if not all experimental models of DLBCL do express IgM (and not IgG).
- 3) Because DLBCL is probably the tumor that has been most comprehensively analyzed within the last years, all data coming from the study basically confirm previous studies. In clear contrast, we know little about the composition and function of immune and microenvironment cells surrounding DLBCL cells. Were these transduced cells grown in immunocompetent mice, for instance the most aggressive BCL6+BCL2+P53^{kd}+CCND3 cells? This would be extraordinary, in order to advance towards defining immune cell features during DLBCL development
- 4) Did the authors investigate the activation of NF- κ B signaling in the tumors? Some activating mutations (CD79a, CARD11) were identified in the screening (fig 3c). Did the authors genetically activate NF- κ B signaling in the GC cells (i.e. CD79a/b, A20 deletion)? It could be scientifically relevant, as virtually all ABC DLBCLs and many GCB DLBCLs exhibit functional activation of this pathway.
- 5) Did authors perform pre-clinical trials in the mice transplanted with the DLBCL cells, testing drugs in clinical practice (rituximab, CHOP, venetoclax, etc)?
- 6) I am curious about whether the authors achieved plasmacytic differentiation of the GC cells in the model, with and without oncogenic activation. They are surely aware that experimental models of plasma cell tumors (multiple myeloma) would be extremely interesting

Reviewer #2 (Remarks to the Author):

The manuscript by Caeser and colleagues represents a significant step forward and provides a new avenue away from the dependency on cell lines, the mainstay of functional studies, along with mouse models to test or identify the effects of genes implicated in B cell lymphomagenesis.

I will take each of the primary results sections in turn for discussion:

For completeness, it is worth (within the supplemental data) discussing the YK6 culture model and other iterations tested as part of finalising the culture strategy; this will benefit other groups, who want to set up this or related models in their own laboratories. Regarding the source of GC B cell, is there a difference between paediatric and adult 'GC' cells.

I think the backbone of forced expression of BCL2, BCL6, MYC deserves some further commentary; how do the levels of these proteins compare with malignancy and is there significant heterogeneity in relationship to their expression across different experiments. What proportion of patients with DLBCL do these different backbones represent? How similar are primary DH DLBCL profiles with these transduced counterparts.

I wondered if the authors have any data on DNA methylation and if acquired events copy those observed in primary lymphomas?

I also feel the authors need to discuss/discriminate the notion between cancer initiating events and those that may drive disease progression; I agree that its welcome to see TP53, CDKN2A and PTEN emerging, but equally most of the epigenetic mutations/silencing are not detected/enriched; so are there certain types of mutation/event that will be best captured using this model. How variable will the readout be and is it all dependent on the backbone utilised.

The GNA13 and the downstream data is reasonable compelling but given the 700 or so targets tested it would be useful to have a sense of the level of enrichment of 'interesting targets' that appear to be impacting biology.

Authors' response to the Editor and to the Reviewers.

We are grateful to the Editor and to the reviewers for their insightful comments on our manuscript. On the basis of these comments we have made considerable improvement to the manuscript including the addition of new RNA-Seq and CRISPR screening data, as well as additional discussion of existing results.

We have responded to each point individually below, with our responses in blue text.

Reviewer #1 (Remarks to the Author):

This manuscript by Caeser and colleagues reports the generation of an innovative model system to grow and expand human tonsillar GC B lymphocytes in the FDC-like feeder line HK genetically modified to express Cd40lg and to secrete IL21. Remarkably, transduction of these primary cells with two different combinations of lymphoma-associated oncogenes (MYC+BCL2 or BCL6+BCL2) led to in vitro malignant transformation. These tumors recapitulated phenotypical and gene expression features of human GC-derived lymphomas. High-throughput CRISPR-based screening using gRNAs targeting 692 genes mutated or deleted in lymphoma revealed well-known TSGs (P53, PTEN, CDK2A) and the GNA13 gene, also commonly mutated in GC-derived DLBCL (and of other downstream proteins S1PR2 and P2RY8), which collectively confirms the validity of the system to investigate GC lymphomagenesis. Transduced cells were capable to grow in immunodeficient mice, and retained the principal features of GCB and ABC DLBCLs.

Overall, this is an interesting, outstanding technical manuscript that reports for the first time (as far as I know) the oncogenic transformation of human primary GC B lymphocytes *ex vivo* upon transduction of known oncogenic drivers. This extraordinary, reliable, simple setting will allow the scientific community not only to continue investigating GC-derived lymphoma biology and development, but also testing drug combinations pre-clinically.

We thank the reviewer for recognizing the significance of our findings, which represent the first report of the transformation of human germinal center B cells into synthetic *in vivo* models of DLBCL. We agree this will provide the scientific community with a valuable tool to investigate the genetics of GC-derived lymphoma and will also serve as a model for preclinical drug testing.

A number of doubts, observations and suggestions follow:

1) Regarding DLBCLs developed in vitro and in mice, were these classified as ABC or GCB subtypes and were these subtypes correlated with specific mutation combinations?

To address the reviewer's question, we have now performed RNA sequencing on sixteen "synthetic" tumors derived from human GC B cells transduced with different combinations of oncogenes and engrafted into immunodeficient mice. We compared their transcriptional profiles to publicly available RNA-Seq data from DLBCL patients enrolled in the GOYA Trial¹, as well as published RNA-Seq from Burkitt Lymphoma patients². Sequencing data from these public data sets were reanalyzed using our RNA-seq pipeline and adjusted for batch effect to allow comparison with our synthetic tumors. GOYA samples were annotated by NanoString technology at source as ABC, GCB or unclassified. As a further

comparison we also sequenced six well-established lymphoma cell lines including two from each of ABC, GCB and BL.

Principal component analysis applied to expression of 858 genes³ that differentiate between ABC vs GCB DLBCL revealed that whilst some synthetic tumors cluster closely with ABC DLBCL, others sit in a position intermediate between GCB DLBCL and Burkitt lymphoma. This is perhaps not surprising given the forced expression of MYC in these tumors. In terms of whether individual oncogenes push cells to being GCB / ABC classification, it is notable that the four synthetic tumors located in the most GCB-like region of the PCA plot were those where the oncogenic retroviral backbone included *BCL6*. Of note, our synthetic tumors show greater similarity to DLBCL than do the cell lines. This data is now included as Figure 6a.

We also compared two recently described signatures associated with double hit (*BCL2* / *MYC* translocated) lymphoma^{4,5}. Both signatures showed clear enrichment suggesting that the particular oncogenic backbone we used for the creation of these tumors (including *MYC* and *BCL2*) generated tumor models with a gene expression profile that most closely approximates MYC/BCL2 double hit lymphoma. This data is now included as Figure 6b.

We anticipate that future studies will examine in greater depth the ability to tailor the oncogenic backbone to best model distinct subtypes of DLBCL.

2) While signs of IGVH somatic hypermutation are shown in the tumors, did class-switch recombination occur? Did tumor cells express surface IgM? Did the authors stain for IgG expression? This could be clinically relevant, as half of DLBCLs in patients show class-switched phenotypes, but most if not all experimental models of DLBCL do express IgM (and not IgG).

We have addressed the reviewer's question using flow cytometry and RNA-Seq data.

It is important to note that our culture experiments started with purified germinal center B cells, many of which had already switched to expression of surface IgG. When kept in culture, IgG expressing cells became the dominant population over time (Supplementary Figure 7d). Low expression of surface IgM was seen in some cells at the start of culture but was gradually lost over time (Supplementary Figure 7d).

We found it more challenging to determine BCR isotype on synthetic tumors by either flow cytometry or immunohistochemistry. Such difficulty in accurately detecting heavy chain isotype in GC cells has been reported previously⁶. However, the RNA-Seq performed on tumors allowed us to comment on the relative transcription of each immunoglobulin isotype. Lymphoma cell lines of known isotype were included as controls. As expected, we observed exclusive expression of the IgM transcripts in ABC DLBCL lines and BL lines. In contrast, only IgG transcripts were detected in the two GCB lines tested. Primary, *ex vivo* GC B cells showed expression of IgM, G and A transcripts, a pattern that was also seen in cultured GC B cells at 73 days. Analysis of synthetic tumors showed varied expression of the constant genes across different tumors. Notably, there was clear expression in many tumors of IgG and IgA transcripts. This data is now included as Supplementary Figure 7e. However, we remain cautious in drawing firm conclusions from these transcriptional data over class switch recombination as many cells were already switched at the outset of the experiment. Future studies may be able to address this question by selecting exclusively surface IgM positive cells at the outset of the experiment.

3) Because DLBCL is probably the tumor that has been most comprehensively analyzed within the last years, all data coming from the study basically confirm previous studies. In clear contrast, we know little about the composition and function of immune and microenvironment cells surrounding DLBCL cells. Were these transduced cells grown in immunocompetent mice, for instance the most aggressive BCL6+BCL2+P53^{dd}+CCND3 cells? This would be extraordinary, in order to advance towards defining immune cell features during DLBCL development

We have not tried to implant transduced GC B cells into immunocompetent mice. Whilst the more aggressive combinations of oncogenes form synthetic tumors rapidly in immunodeficient mice, we consider it extremely unlikely that cross-species cells would be able to engraft in an immunocompetent mouse. For instance, established lymphoma cell lines, which form tumors even faster in immunodeficient mice, are unable to engraft in immunocompetent mice. This is predominantly because of rapid rejection of mismatched MHC by host CD8 T cells. However, we fully accept the reviewer's point regarding the increasing importance of understanding the interaction of the microenvironment with DLBCL cells and the possibility that the experimental system we describe here might, in future studies, provide a tool to experimentally manipulate the expression of surface molecules that regulate the tumor-host immune interaction. These experiments may be best performed in the future by employing humanized immune system mice

4) Did the authors investigate the activation of NF- κ B signaling in the tumors? Some activating mutations (CD79a, CARD11) were identified in the screening (fig 3c). Did the authors genetically activate NF- κ B signaling in the GC cells (i.e. CD79a/b, A20 deletion)? It could be scientifically relevant, as virtually all ABC DLBCLs and many GCB DLBCLs exhibit functional activation of this pathway.

To address these questions we first used our RNA-Seq data to investigate NF- κ B activity in both cultured cells and in synthetic tumors. A heat map showing mRNA expression of a combined NF- κ B signature (Supplementary Table 4) is now shown in Supplementary Figure 7a. This reveals strong NF- κ B activity in the cultured B cells, which we attribute to the CD40 stimulation provided from the feeder cells (YK6-CD40lg-IL21). However, NF- κ B activity is weaker in the synthetic tumors, in which CD40 stimulation is not present. This result is expected given the specific combinations of oncogenes used to generate these particular tumors. Importantly, as the reviewer has mentioned, our CRISPR screen reveals ongoing dependency of cultured / transduced GC B cells on genes promoting NF- κ B signaling as seen upon deletion of *CARD11* and *CD79B*.

We looked specifically at the effect of depleting A20, a negative regulator of NF- κ B, and observed an associated increased competitive fitness (Supplementary Figure 2c). Overall these data suggest that NF- κ B activity can be experimentally manipulated in our system, both by genetic methods and by altered exogenous stimuli. Altered NF- κ B signaling output appears to affect competitive fitness of cells, suggesting the system will be a useful tool to model the genetic contributors to NF- κ B activity in B cell lymphoma.

5) Did authors perform pre-clinical trials in the mice transplanted with the DLBCL cells, testing drugs in clinical practice (rituximab, CHOP, venetoclax, etc)?

We have not performed pre-clinical trials using our transplanted mice and consider this to be beyond the scope of this initial manuscript. However, we envisage future studies where the effect of potential therapeutic agents could be tested on the background of specific genetic alterations.

6) I am curious about whether the authors achieved plasmacytic differentiation of the GC cells in the model, with and without oncogenic activation. They are surely aware that experimental models of plasma cell tumors (multiple myeloma) would be extremely interesting

We did not observe expression of the plasma cell marker CD138 in our cultured cells (Supplementary Figure 1f) or our synthetic tumors (Figure 5c). We performed an ELISA to examine IgM and IgG secretion in three cell lines and eight synthetic tumors. None of the synthetic tumors secreted detectable levels of IgM or IgG suggesting these tumors had not undergone plasmacytic differentiation. This can be seen in the Figure below.

Our culture system included continuous exposure to CD40ligand, which has previously shown to block human plasma cell differentiation⁷. Furthermore, the forced expression of BCL6 would be expected to block plasma cell differentiation. These conditions were selected to optimally model DLBCL, a core biological component of which includes blocked plasma cell differentiation. As part of a separate initiative, we are investigating whether the culture system can be adapted to allow expansion and transduction of human B cells with different oncogene backgrounds and altered growth conditions such as IL-10 to promote plasma cell differentiation as a model system to study myeloma genetics.

Reviewer #2 (Remarks to the Author):

The manuscript by Caeser and colleagues represents a significant step forward and provides a new avenue away from the dependency on cell lines, the mainstay of functional studies, along with mouse models to test or identify the effects of genes implicated in B cell lymphomagenesis.

I will take each of the primary results sections in turn for discussion:

For completeness, it is worth (within the supplemental data) discussing the YK6 culture model and other

iterations tested as part of finalising the culture strategy; this will benefit other groups, who want to set up this or related models in their own laboratories. Regarding the source of GC B cell, is there a difference between paediatric and adult 'GC' cells.

The development of the HK / YK6 feeder system has been previously described by Kim et al⁸. The use of *TERT* / *CDK4* / P53dd is described in the methods. A representative experiment comparing different methods of stimulating B cells is now shown (Supplementary Figure 1b). This shows the superiority of membrane based CD40lg stimulation over soluble CD40 ligation and the enhanced proliferation of IL21 stimulated cells. We anticipate that in the context of CD40lg stimulation, other cytokines may also allow effective expansion of GC B cells as has been previously described⁹ however, the system described in this manuscript was developed to maximize efficiency of retroviral and lentiviral transduction.

We agree it would be interesting to compare adult and pediatric GC cells. However, whilst pediatric tonsils are routinely removed in clinical practice and represent a ready source of GC B cells, such cells are not readily available from healthy adults. These logistical difficulties meant our access to normal GC B cells was restricted to pediatric samples. Additionally, we considered it important to exclude the possibility that latent EBV infection might contribute to B cell immortalization in our experiments. We therefore excluded GC B cells where we found evidence of latent EBV infection (described in methods "Purification of human Germinal Center B cells"). This was a rare finding in tonsils derived from pre-school children but prior EBV exposure, and persistent latent EBV infection, is likely to be much more common in an adult population.

I think the backbone of forced expression of BCL2, BCL6, MYC deserves some further commentary; how do the levels of these proteins compare with malignancy and is there significant heterogeneity in relationship to their expression across different experiments. What proportion of patients with DLBCL do these different backbones represent? How similar are primary DH DLBCL profiles with these transduced counterparts.

BCL2 / *BCL6* / *MYC* were chosen as the oncogenic backbone as they are amongst the most frequently expressed oncogenes in human DLBCL. The proportion of patients expressing these proteins was previously determined¹⁰ using immunohistochemistry on biopsy material from 279 ABC and GCB DLBCL patients enrolled in the RICOVER-60 trial. BCL2/BCL6/MYC expression was observed in 78%/78%/37% of ABC DLBCL and 50%/94%/29% of GCB DLBCL¹⁰. This data, adapted from Staiger et al. is now shown in Supplementary Figure 4a. Therefore, we consider these genes to be an entirely appropriate backbone to use in our synthetic tumors. Of course, we anticipate that other oncogenic combinations will be investigated as part of future studies.

To confirm protein expression of BCL6, BCL2 and MYC from each of the constructs used, we performed Western blot on transduced human GC B cells cultured for 6 days, as well as in synthetic tumors. This data is now included in Supplementary Figure 6a. We also used RNA-Seq to compare expression in our synthetic tumors with that of existing RNA-Seq from primary ABC DLBCL, GCB DLBCL and Burkitt lymphoma biopsies, as well as established lymphoma cell lines. As expected, considerable variability of expression was seen between human tumors. Our synthetic tumors tended to have higher expression of each of the forced oncogenes, and less variability. This might suggest that an optimum expression level is selected for within each oncogenic combination. This data is now shown in Supplementary Figure 6b.

To investigate the similarity with DH DLBCL we compared RNA-Seq data from our synthetic tumors with two recently described gene expression signatures related to double hit lymphoma^{4,5}. As discussed in response to Reviewer 1 above, we see enrichment for both signatures suggesting that synthetic tumors based on a *MYC/BCL2* background may serve as a useful tool to model double hit lymphoma.

I wondered if the authors have any data on DNA methylation and if acquired events copy those observed in primary lymphomas?

We do not have data on DNA methylation in these tumors but agree this would be interesting to perform as part of a future study.

I also feel the authors need to discuss/discriminate the notion between cancer initiating events and those that may drive disease progression; I agree that its welcome to see TP53, CDKN2A and PTEN emerging, but equally most of the epigenetic mutations/silencing are not detected/enriched; so are there certain types of mutation/event that will be best captured using this model.

We agree these are very important points. As the reviewer points out, notable absentees from the genes enriching in our CRISPR screens were those encoding the histone modifiers *CREBBP*, *EP300* and *KMT2D*. These genes show very frequent inactivating mutations in DLBCL and follicular lymphoma, where mutations are almost always clonal suggesting that they arise at an early stage of lymphomagenesis¹¹⁻¹⁴, potentially before the germinal center stage. Interestingly, mouse models of *Crebbp* and *Kmt2d* knockout show a more pronounced tumor-promoting phenotype when Cre-mediated knockout is induced at earlier stages of B cell development, prior to the germinal center¹⁵⁻²⁰. Therefore, one potential explanation for why these genes do not enrich in our screens (performed using mature GC B cells), is that during lymphomagenesis, the predominant biological effect of these mutations is exerted prior to the germinal center stage. In contrast, the mutant genes enriched in our screens may reflect those that have the greatest effect in a GC B cell. Interestingly, some of the top hits from our screen show similarity to those described as being associated with transformation of follicular lymphoma into high grade lymphoma (*GNA13*, *CDKN2A*, *TP53*, *P2RY8*, *S1PR2*)¹³. We speculate that this might be consistent with our screen detecting those mutations that provide proliferation or survival advantage to an already "corrupted" GC B cell. In developing lymphoma cells, this "corruption" might reflect pre-GC mutations of *CREBBP*, whilst in our screen, this oncogenic corruption could be provided already by the forced expression of *BCL2* / *BCL6*. Text of this discussion is now included in the manuscript.

How variable will the readout be and is it all dependent on the backbone utilized?

The enrichment of genes across experiments performed in three separate human donors was remarkably consistent; four genes (*GNA13*, *TP53*, *CDKN2A*, *ATRX*) featured in the top ten most enriched genes in all three donors. However, we expect that the oncogenic backbone used will have a major influence on the co-operating genes selected. Indeed, we consider the ability to vary the backbone to be one of the key advantages of the system. To test this assumption, we performed a further CRISPR screen, this time using *BCL2* and *MYC* as the oncogenic backbone (This data is now shown in Supplementary Figure 3b). Interestingly, in the absence of *BCL6* expression, we no longer see strong enrichment for depletion of *GNA13*. This fits with the distribution of *GNA13* mutations in human DLBCL, which are found predominantly in the EZB subtype³, which expresses the highest levels

of BCL6. Interestingly, some of the most enriched gRNAs in the context of MYC expression appear to target genes encoding members of the ZFP36 (Tristetraprolin) family of RNA binding proteins. This finding is consistent with existing biological knowledge of these proteins, which negatively regulate cell cycle²¹ and have been demonstrated to oppose cellular transformation in a mouse model of MYC-induced lymphoma²².

Therefore, we conclude that the genes enriched in these screens are heavily influenced by the oncogenic backbone used. We envisage that future screens may vary the oncogenic backbone in order to identify synergistic combinations of genetic alterations and model different genetic subtypes. We also envisage future screens that vary the stimulation cocktail, perhaps looking for genetic alterations that relieve dependency on either CD40lg or cytokine signaling.

The GNA13 and the downstream data is reasonable compelling but given the 700 or so targets tested it would be useful to have a sense of the level of enrichment of 'interesting targets' that appear to be impacting biology.

GNA13 and *TP53* were by far the most enriched hits from the CRISPR screening experiments. Indeed, in one screen, the nine gRNAs targeting *GNA13* collectively represented 79% of all reads by day70, shown graphically in Supplementary Figure 3a. This extremely potent expansion may serve to mask to some degree the potential expansion of gRNAs that target weaker tumor suppressors. However, we agree with the reviewer that it is important to discuss in more depth the top scoring genes in our screens. To this end we have now included summary data from three lymphoma-focused CRISPR screens, performed in *BCL2-BCL6* transduced human GC B cells from three separate tonsil donors. CRISPR-gene scores were calculated for each screen as described in the methods section and genes then ranked based on average CRISPR-gene score across all three screens. The twelve most enriched genes were *TP53*, *GNA13*, *CDKN2A*, *ATRX*, *NFKBIA*, *ZFP36L1*, *ZNF281*, *PTEN*, *FBXO11*, *FUBP1*, *S1PR2* & *NFKBIE*. The majority (those highlighted) of these are associated with a tumor suppressor function in lymphoma established in the literature^{3,21,23,24}, either from evidence of recurrent genetic inactivation or from their ability to inhibit cancer-promoting pathways. The next 24 most enriched genes included *TET2*, *TSC1*, *GSK3B*, *RB1*, *CDKN2B*, *P2RY8* and *SOCS1*, also implicated as tumor suppressor genes. Thus, the most enriched genes contained a predominance of recognized tumor suppressor genes.

Although our study was not designed for the sensitive detection of drop-outs, the most depleted gRNAs were enriched for those that targeted established oncogenes. Indeed, the two most depleted gRNAs were those targeting *POU2AF1* and *MYC*, both well-established oncogenes in B cell lymphoma.

We have added this data to the manuscript as Figure 3d and the complete data from all three screens is now included as a summary Supplementary Table 2. Further description has been included in the results and discussion sections.

References

1. McCord R, Bolen CR, Koeppen H, et al. PD-L1 and tumor-associated macrophages in de novo DLBCL. *Blood Adv.* 2019;3(4):531-540.
2. Schmitz R, Young RM, Ceribelli M, et al. Burkitt lymphoma pathogenesis and therapeutic targets from structural and functional genomics. *Nature.* 2012;490(7418):116-120.
3. Schmitz R, Wright GW, Huang DW, et al. Genetics and Pathogenesis of Diffuse Large B-Cell Lymphoma. *N Engl J Med.* 2018;378(15):1396-1407.
4. Sha C, Barrans S, Cucco F, et al. Molecular High-Grade B-Cell Lymphoma: Defining a Poor-Risk Group That Requires Different Approaches to Therapy. *J Clin Oncol.* 2019;37(3):202-212.
5. Ennishi D, Jiang A, Boyle M, et al. Double-Hit Gene Expression Signature Defines a Distinct Subgroup of Germinal Center B-Cell-Like Diffuse Large B-Cell Lymphoma. *J Clin Oncol.* 2019;37(3):190-201.
6. Grier DD, Al-Quran SZ, Cardona DM, Li Y, Braylan RC. Flow cytometric analysis of immunoglobulin heavy chain expression in B-cell lymphoma and reactive lymphoid hyperplasia. *Int J Clin Exp Pathol.* 2012;5(2):110-118.
7. Arpin C, Dechanet J, Van Kooten C, et al. Generation of memory B cells and plasma cells in vitro. *Science.* 1995;268(5211):720-722.
8. Kim HS, Zhang X, Klyushnenkova E, Choi YS. Stimulation of germinal center B lymphocyte proliferation by an FDC-like cell line, HK. *J Immunol.* 1995;155(3):1101-1109.
9. Arpin C, Dechanet J, Van Kooten C, et al. Generation of memory B cells and plasma cells in vitro. *Science.* 1995;268(5211):720-722.
10. Staiger AM, Ziepert M, Horn H, et al. Clinical Impact of the Cell-of-Origin Classification and the MYC/ BCL2 Dual Expresser Status in Diffuse Large B-Cell Lymphoma Treated Within Prospective Clinical Trials of the German High-Grade Non-Hodgkin's Lymphoma Study Group. *J Clin Oncol.* 2017;35(22):2515-2526.
11. Pasqualucci L, Khiabani H, Fangazio M, et al. Genetics of follicular lymphoma transformation. *Cell Rep.* 2014;6(1):130-140.
12. Okosun J, Bodor C, Wang J, et al. Integrated genomic analysis identifies recurrent mutations and evolution patterns driving the initiation and progression of follicular lymphoma. *Nat Genet.* 2014;46(2):176-181.
13. Kridel R, Chan FC, Mottok A, et al. Histological Transformation and Progression in Follicular Lymphoma: A Clonal Evolution Study. *PLoS Med.* 2016;13(12):e1002197.
14. Green MR, Gentles AJ, Nair RV, et al. Hierarchy in somatic mutations arising during genomic evolution and progression of follicular lymphoma. *Blood.* 2013;121(9):1604-1611.
15. Jiang Y, Ortega-Molina A, Geng H, et al. CREBBP Inactivation Promotes the Development of HDAC3-Dependent Lymphomas. *Cancer Discov.* 2017;7(1):38-53.
16. Zhang J, Vlasevska S, Wells VA, et al. The CREBBP Acetyltransferase Is a Haploinsufficient Tumor Suppressor in B-cell Lymphoma. *Cancer Discov.* 2017;7(3):322-337.
17. Garcia-Ramirez I, Tadros S, Gonzalez-Herrero I, et al. Crebbp loss cooperates with Bcl2 overexpression to promote lymphoma in mice. *Blood.* 2017;129(19):2645-2656.

18. Horton SJ, Giotopoulos G, Yun H, et al. Early loss of Crebbp confers malignant stem cell properties on lymphoid progenitors. *Nat Cell Biol.* 2017;19(9):1093-1104.
19. Zhang J, Dominguez-Sola D, Hussein S, et al. Disruption of KMT2D perturbs germinal center B cell development and promotes lymphomagenesis. *Nat Med.* 2015;21(10):1190-1198.
20. Ortega-Molina A, Boss IW, Canela A, et al. The histone lysine methyltransferase KMT2D sustains a gene expression program that represses B cell lymphoma development. *Nat Med.* 2015;21(10):1199-1208.
21. Galloway A, Saveliev A, Lukasiak S, et al. RNA-binding proteins ZFP36L1 and ZFP36L2 promote cell quiescence. *Science.* 2016;352(6284):453-459.
22. Rounbehler RJ, Fallahi M, Yang C, et al. Tristetraprolin impairs myc-induced lymphoma and abolishes the malignant state. *Cell.* 2012;150(3):563-574.
23. Shaffer AL, 3rd, Young RM, Staudt LM. Pathogenesis of human B cell lymphomas. *Annu Rev Immunol.* 2012;30:565-610.
24. Muppidi JR, Schmitz R, Green JA, et al. Loss of signalling via Galpha13 in germinal centre B-cell-derived lymphoma. *Nature.* 2014;516(7530):254-258.

REVIEWERS' COMMENTS:

Reviewer #2 (Remarks to the Author):

The authors have addressed all my comments well and indeed included several new data/experiments which has further strengthened their manuscript.

Reviewer #2 (Remarks to the Author): The authors have addressed all my comments well and indeed included several new data/experiments which has further strengthened their manuscript.

We thank the reviewer for his support of publication.